

# Ozone source apportionment during peak summer events over southwestern Europe

María Teresa Pay[1], Gotzon Gangoiti[2], Marc Guevara[1], Sergey Napelenok[3], Xavier Querol[4], Oriol Jorba[1], Carlos Pérez García-Pando[1]

[1]Earth Sciences Department, Barcelona Supercomputing Center, BSC, c/Jordi Girona, 29, 08034 Barcelona, Spain
[2]Department of Chemical and Environmental Engineering, University of the Basque Country UPV/EHU, ETSI-Bilbao School of Engineering, Alda. de Urquijo s/n, E-48013 Bilbao, Spain.
[3]U.S. EPA, Research Triangle Park, NC, USA.
[4]Institute of Environmental Assessment and Water Research, IDAEA-CSIC, c/Jordi Girona, 18-26, 08034 Barcelona, Spain.

*Correspondence to*: María Teresa Pay (maria.pay@bsc.es)

**Abstract.** It is well established that in Europe, high $O_3$ concentrations are most pronounced in southern/Mediterranean countries due to the more favorable climatological conditions for its formation. However, the contribution of the different sources of precursors to $O_3$ formation within each country relative to the imported (regional and hemispheric) $O_3$ is poorly quantified. This lack of quantitative knowledge prevents local authorities from effectively designing plans that reduce the exceedances of the $O_3$ Target Value set by the European Air Quality Directive. $O_3$ source attribution is a challenge because the concentration at each location and time results not only from local biogenic and anthropogenic precursors, but also from the transport of $O_3$ and precursors from neighbouring regions, $O_3$ regional and hemispheric transport and stratospheric $O_3$ injections. Our study applies and thoroughly evaluates a countrywide $O_3$ source apportionment method implemented in a Chemical Transport Model (CTM) at high resolution (4 x 4 km) over the Iberian Peninsula (IP) to understand and quantify the origin of peak $O_3$ events over a 10-day period covering the most frequent synoptic summer conditions in the IP. The method tags both $O_3$ and its gas precursor emissions from source sectors within one simulation and each tagged species is subject to the typical physical processes (advection, vertical mixing, deposition, emission and chemistry) as the actual conditions remain unperturbed. We quantify the individual contributions of the largest $NO_x$ local sources to high $O_3$ concentrations compared to the contribution of imported $O_3$. We show for the first time that imported $O_3$ is the largest input to the ground-level $O_3$ concentration in the IP, accounting for 46% to 68 % of the daily mean $O_3$ concentration during exceedances of the European Target Value. The hourly imported $O_3$ increases during typical northwestern advections (70-90%, 60-80 µg/m³), and decreases during typical stagnant conditions (30-40%, 30-60 µg/m³) due to the local NO titration effect. During stagnant conditions, the anthropogenic precursors control the $O_3$ peaks in areas downwind of the main urban and industrial regions (up to 40% in hourly peaks). We also show that ground-level $O_3$ concentrations are strongly affected by vertical mixing of $O_3$-rich layers present in the free troposphere, which result from local/regional layering and accumulation, and continental/hemispheric transport. Indeed, vertical mixing largely explains the presence of imported $O_3$ at ground level in the Iberian Peninsula. Our results demonstrate the need for detailed quantification of the local and remote



contributions to high $O_3$ concentrations for local $O_3$ management, being the $O_3$ source apportionment an essential analysis prior to the design of $O_3$ mitigation plans in any non-attainment area. To achieve the European $O_3$ objectives in southern Europe, *ad hoc* local actions should be complemented by decided national and European-wide strategies.

## 1 Introduction

Tropospheric ozone ($O_3$) is an air pollutant of major public concern as it harms human health (WHO, 2013) and sensitive vegetation (Booker et al., 2009), and contributes to climate change (Jacob and Winner, 2009). $O_3$ is formed in the atmosphere through nonlinear photochemical reactions among carbon monoxide (CO), peroxy radicals generated by the photochemical oxidation of volatile organic compounds (VOC), and nitrogen oxides ($NO_x$) (Crutzen, 1973). Therefore, meteorological stagnation, high temperature, and low precipitation enhance tropospheric $O_3$ formation (Demuzere et al.,

2009; Otero et al., 2016). Atmospheric circulation also controls the short and long-range transport of $O_3$ affecting its lifetime in the atmosphere (Monks et al., 2015). For example, the transport of precursors emitted in urban and industrialized areas may cause $O_3$ production downwind (Holloway et al., 2003).

According to the European Environmental Agency (EEA) around 95-98% of the population in Europe during 2013-2015

were exposed to $O_3$ concentrations that exceeded the guidelines of the World Health Organization (WHO) (EEA, 2017). These guidelines establish a maximum daily 8-hour averaged (MDA8) $O_3$ concentration of 100 $\mu gm^{-3}$ never to be exceeded. The European Air Quality Directive (2008/50/EC) is less restrictive as it sets an $O_3$ Target Value of 120 $\mu gm^{-3}$ for the MDA8 concentration, which can be exceeded up to 25 days per calendar year averaged over three years.

Southern European countries around the Mediterranean Basin are particularly exposed to exceedances of the $O_3$ Target Value in summer due to the influence of frequent anticyclonic and clear-sky conditions that favour photochemical $O_3$ formation in the troposphere (EEA, 2017). In addition, its geographic location also makes the Basin a receptor of the long-range transport of pollution from Europe, Asia and even North America (Lelieveld et al., 2002; Gerasopoulos, 2005). The importance of long-range transport on surface $O_3$ has been studied in the Mediterranean Basin, indicating that the emission

sources within the Basin have a dominating influence on surface $O_3$, while remote sources are more important than local sources for $O_3$ mixing ratios at higher altitudes (Richards et al., 2013; Safieddine et al., 2014). Recent studies suggest that the upper $O_3$-rich air masses could increase the surface $O_3$ concentration in the Mediterranean Basin (Kalabokas et al., 2017; Querol et al., 2018). Further detailed and quantitative studies on the mechanism linking upper $O_3$-rich layer with increases of the ground-level $O_3$ concentration in episodes need further clarification particularly regarding the contribution of $O_3$

transported at regional and hemispheric scales.



Several studies in the Iberian Peninsula have addressed the causes of $O_3$ episodes looking at the circulation of air masses (Millán, 2014, and references therein). In the Atlantic region, the blocking anticyclones over Western Europe favour the inter-regional transport of $O_3$ in the area and its accumulation for several days during the most severe episodes (Alonso et al., 2000; Gangoiti et al., 2002, 2006; Valdenebro et al., 2011; Saavedra et al., 2012). On the other hand, in the Mediterranean coast, the typical summer synoptic meteorological conditions with a lack of strong synoptic advection, combined with the orographic characteristics and the sea and land breezes, favour episodes where high levels of $O_3$ are accumulated by recirculation of air masses loaded with $O_3$ precursors (Millán et al., 1997 and 2000; Toll and Baldasano, 2000; Gangoiti et al., 2001; Pérez et al., 2004; Jiménez et al., 2006; Gonçalves et al., 2009; Millán, 2014, Querol et al., 2017; Querol et al., 2018). The coupling between synoptic and mesoscale processes governing the levels of $O_3$ in the Western Mediterranean Basin need further research in order to understand the $O_3$ intercontinental contribution. Furthermore, from our understanding there is a lack of research quantifying the contribution of the activity sources to the $O_3$ local formation during peak events in this region.

$O_3$ analyses in the Western Mediterranean Basis show that regional background $O_3$ levels have remained high without significant changes (EEA, 2016; EMEP-CCC, 2016; Querol et al., 2016). However, they have increased at traffic and urban background sites (EEA, 2016; Querol et al., 2016; Sicard et al., 2016; Saiz-Lopez et al., 2017). The reasons behind the urban $O_3$ upward trend are not clear yet; it might be a result of the reduction of NO emissions relative to $NO_2$ and therefore a lower NO titration effect or the fact that urban $O_3$ formation is VOC-limited, and a reduction in $NO_x$ emissions might enhance $O_3$ formation. The most intense $O_3$ events in the last decade, measured by the number of exceedances of the $O_3$ Target Value are recorded over areas downwind of large urban and industrial hot spots (Monterio et al, 2012; Querol et al., 2016; EEA, 2016). Overall, the number of these type $O_3$ events occur in June-July and during summer heatwaves (i.e., 2003 and 2015).

According to the European Air Quality Directive, in zones exceeding the $O_3$ Target Value, Member States must develop plans to attain compliance by reducing the emission of $O_3$ precursors. Abatement of tropospheric $O_3$ concentration in the Western Mediterranean Basin has been insufficient so far (Querol et al., 2018). Effective planning requires an accurate quantitative knowledge of the sources of these precursors and their respective contributions to the exceedances of the $O_3$ Target Value (Querol et al., 2016; Borrego et al., 2015). However, source attribution of surface $O_3$ concentration remains a challenge, because the concentration at each location and time results not only from local biogenic and anthropogenic precursors, but also from the transport of $O_3$ and its precursors from neighbouring regions, $O_3$ hemispheric transport (UNECE, 2010), and stratospheric $O_3$ injections (Monks et al., 2015).

At present, there are no methods based on observations that distinguish the origin of $O_3$. Despite their inherent uncertainties, Chemical Transport Models (CTMs) allow apportioning the contribution of any source (by sector and/or region) to $O_3$ concentrations. The most widely used approach is the "brute force" method, which consists on running an ensemble of



simulations zeroing out the sources one by one and then comparing them with a baseline simulation that accounts for all of the sources. Several $O_3$ source apportionment studies at European scale have applied the brute force method to quantify the contribution of one or two emission sectors. For example, road transport emissions with the EMEP model (Reis et al., 2000), biogenic and anthropogenic emissions with the Polyphemus model (Sartelet et al. 2012), transport-related emissions

including road transport, shipping, and aviation with the WRF-CMAQ model (TRANSPHORM, 2014), and ship emissions with CAMx (Aksoyuglu et al., 2016). This approach is simple to implement, as it does not require additional coding in the CTM. However, as it quantifies the contribution of each source based on its absence, it does not reproduce actual atmospheric conditions, and therefore it is susceptible to inaccuracies in the prediction of $O_3$ peaks under non-linear regimes (Cohan and Napelenok, 2011). Despite these limitations, brute force is useful for analysing the concentration responses to

emission abatement scenarios.

Recently, CTMs include algorithms that tag multiple pollutants by source (region and/or sector) all the way through the pollutant's lifetime, from emission to deposition. This integrated source apportionment approach has several advantages. First, it allows identifying the main sources contributing to high $O_3$ levels under actual atmospheric conditions, which is a

preliminary step towards designing refined and efficient emission abatement scenarios. Second, as we show below, it supports enhanced model evaluation and therefore potential model improvements by identifying problems in emission estimates (sectors or regions) or chemical boundary conditions. The Integrated Source Apportionment Method (ISAM) within the Community Multiscale Air Quality (CMAQ) model has shown promising results for $O_3$ tagging, exhibiting less noise in locations where brute force results are demonstrably inaccurate (Kwok et al., 2013, 2015). Recent ISAM

experiments have quantified that the contribution of traffic in the cities of Madrid and Barcelona to the daily $O_3$ peaks downwind of the urban areas is particularly significant (up to 80-100 $\mu gm^{-3}$) (Valverde et al., 2016a). $O_3$ tagging methods are also included in other regional and global models applied over Europe (Karamchandani et al., 2017; Butler et al., 2018).

The integrated source apportionment tools combined with high-resolution emission and meteorological models can help

unravelling the sources responsible for peak summer events of $O_3$ in the Western Mediterranean Basin. Quantifying the contribution of emission sources during acute $O_3$ episodes is a prerequisite for the design of future mitigation strategies in the region. Our study applies for the first time a countrywide $O_3$ source apportionment at high resolution over the Iberian Peninsula to investigate the local sources responsible for high $O_3$ concentration compared to the imported (regional and hemispheric) $O_3$ during a typical summer episode. Our analysis focuses on the period between July 21[st] and 31[st], 2012, which

is representative of the typical summer synoptic conditions in the region. We use the CMAQ-ISAM within the CALIOPE air quality forecast system for Spain (www.bsc.es/caliope), which runs at 4-km horizontal resolution over the IP. The system is fed by the HERMESv2.0 emission model, which provides disaggregated emissions based on local information and state-of-the-art bottom-up approaches for the most polluting sectors.



The paper is organized as follows. In Section 2 we introduce the CALIOPE system, the set-up of ISAM and the HERMESv2.0 emission model for $O_3$ source apportionment studies, and the methodology used to quantify evaluate the model. In Section 3 we demonstrate the representativeness of the selected episode, we evaluate the model, and we provide an analysis of the source-sector contribution to the Spanish $O_3$ under the different synoptic patterns occurring during the study

period. In Section 4, we discuss our findings, the regulatory implications, and future research.

## 2 Methodology

### 2.1 Air quality model

We used the CALIOPE air quality modelling system (www.bsc.es/caliope) to simulate the $O_3$ dynamics over the IP during the selected episode. CALIOPE is described elsewhere (Baldasano et al., 2008; Pay et al., 2014; Valverde et al., 2016a; and

reference therein). The system consists of the HERMESv2.0 emission model (Guevara et al, 2013), the WRF-ARWv3.6 meteorological model (Skamarock and Klemp, 2008), the CMAQ v5.0.2 chemical transport model (Byun and Schere, 2006) and the BSC-DREAM8bv2 mineral dust model (Basart et al., 2012). CALIOPE first runs over Europe at 12-km resolution (EU12 domain) and then over the IP at 4-km (IP4 domain) (Fig. S1). In the present work, the system is configured with 38 sigma layers up to 50 hPa, both for WRF and CMAQ. The planetary boundary layer is characterized with approximately 11

layers, where the bottom layer's depth is ~39 m. The EU12 domain uses meteorological initial and boundary conditions from the Final Analyses provided by the National Centers of Environmental Prediction (FNL/NCEP) at 0.5° by 0.5°. The first 12 h of each meteorological run are treated as cold start, and the next 23 h are provided to the chemical transport model. Boundary conditions for reactive gases and aerosols come from the global MOZART-4/GEOS-5 model at 1.9° by 2.5° (Emmons et al., 2010). CMAQ uses the CB05 gas-phase mechanism with active chlorine chemistry, an updated toluene

mechanism (CB05TUCL; Whitten et al., 2010; Sarwar et al., 2012), and the sixth generation CMAQ aerosol mechanism including sea salt, aqueous/cloud chemistry and the ISORROPIA II thermodynamic equilibrium module (AERO6; Reff et al., 2009; Appel et al., 2013). Table S1 depicts the remaining CALIOPE configuration options.

For the IP4 domain, HERMESv2.0 estimates emissions for Spain with a temporal and spatial resolution of 1 h and up to 1

25    km by 1 km, according to the Selected Nomenclature for Air Pollution (SNAP), which are then aggregated to 4 km resolution (Guevara et al., 2013). HERMESv2.0 is currently based on 2009 data, which is the most recent year with updated information on local emission activities. For neighbouring countries and international shipping activities, HERMESv2.0 uses the annual gridded national emission inventory provided by the European Monitoring and Evaluation Programme (EMEP) disaggregated to 4 km resolution using a SNAP-sector-dependent spatial, temporal and speciation treatment (Ferreira et al.,

2013). The VOC and $NO_x$ emissions from vegetation, which are critical in the formation of $O_3$, are estimated with the Model of Emissions of Gas and Aerosols from Nature (MEGANv2.0.4) (Guenther et al., 2006) using temperature and solar



radiation from the WRF model. HERMESv2.0 is suitable for source apportionment studies thanks to its level of detail in the calculation of the emission fluxes by source (Guevara et al., 2014).

## 2.2 Ozone source apportionment method

We applied ISAM to quantify contributions from different SNAP categories to the surface $O_3$ over the IP. The ISAM $O_3$ tagging method is a mass balance technique that tags both $O_3$ and its gas precursor emissions ($NO_x$ and VOC) from each source sector within one simulation (Kwok et al., 2013, 2015). Each tagged species undertakes typical physical processes (advection, vertical mixing, deposition, emission and chemistry) without perturbing the actual conditions. The $O_3$ rate of change for each tag in any grid cell is calculated as follows (Eq. 1):

$$\frac{dc_{tag}}{dt} = P_{tag} - D\frac{c_{tag}}{\Sigma_{tag}c},$$ (1)

Where $C_{tag}$ represents the $O_3$ concentration related to a tagged source of interest, $P_{tag}$ is the chemical production rate of $O_3$ formed by the precursors emitted for each tag, and D is the total chemical destruction rate of $O_3$ in this grid cell. Different ratios of $NO_x$/VOC cause the formation of $O_3$ in each grid cell, which is controlled either by $NO_x$- or VOC-limited conditions. ISAM uses the ratio $H_2O_2$/$HNO_3$ to determine whether $O_3$ is $NO_x$- or VOC-sensitive (above or below 0.35, respectively) (Zhang et al., 2009). The bulk $O_3$ concentration in each model grid cell ($P_{bulk}$) is equal to the sum of $O_3$ tracers that were produced in either $NO_x$ or VOC-sensitive conditions (Eq. 2),

$$P_{bulk} = \Sigma_{tag}P_{tag} = \Sigma_{tag}P_{tag}^N + \Sigma_{tag}P_{tag}^V,$$ (2)

where $P_{tag}^N$ and $P_{tag}^V$ are the $O_3$ produced under $NO_x$- and VOC-limited conditions, respectively according Eqs. 3 and 4:

$$P_{tag}^{N,new} = P_{tag}^{N,old} + P_{bulk}^{new}\frac{\Sigma_x NO_{x,tag}}{\Sigma_{tag}\Sigma_x NO_{x,tag}},$$ (3)

$$P_{tag}^{V,new} = P_{tag}^{V,old} + P_{bulk}^{new}\frac{\Sigma_y VOC_{y,tag}\times MIR_y}{\Sigma_{tag}\Sigma_y VOC_{y,tag}\times MIR_y},$$ (4)

$NO_{x,tag}$ and $VOC_{j,tag}$ are the concentrations of the $x$ nitrogen and $y$ VOC species in CB05 that participate in the photochemical $O_3$ formation for each source sector tag and grid cell. $MIR_y$ is the maximum incremental reactivity factor of each $y$ species of VOC emitted by each source sector tag, corresponding to the $O_3$ generating potential of each single VOC species (Carter, 1994).



## 2.3 Ozone tagged species

Table 1 summarizes the $O_3$ tagged sources in the present study and Fig. 1a depicts the HERMESv2.0's estimates of the contribution by each SNAP category to the total emissions of $O_3$ precursors in Spain. The largest $NO_x$ sources are road transport (SNAP7; 42%), non-road transport (SNAP8; 19%), manufacturing industries (SNAP34; 16%), and energy production (SNAP1; 16%). VOC are dominated by biogenic sources (SNAP11; 70%) and to a lesser extent by the agricultural sector (SNAP10; 11%), solvent and other product uses (SNAP6; 9%) and road transport (SNAP7; <7%). The selected (tagged) SNAP categories in this study are the energy, industrial, road transport and non-road transport sectors (Fig. 1b), which account for the 92% of the total $NO_x$ emissions in Spain. An additional tracer (OTHR) gathers the remaining emission categories that were not explicitly tracked (i.e., SNAP2, 5, 6, 9, 10 and 11).

In addition to the selected sources, we tracked the contributions of the chemical boundary conditions (BCON) and the initial conditions (ICON). BCON represents both the $O_3$ directly transported through the IP4 domain boundaries and the formation of $O_3$ resulting from precursors that are also transported through the boundaries. BCON $O_3$ comes from the EU12 parent domain, which includes the $O_3$ produced in Europe and the $O_3$ transported at global scale (both tropospheric and stratospheric $O_3$) provided by the MOZART-4/GEOS model (Fig. S1). Hereinafter, we name BCON $O_3$ as the imported $O_3$ to the IP4 domain. Tagging the $O_3$ initial allows quantifying the number of spin-up days to minimize the impact of model initialization. For the present run, we required 6 days of spin-up to set the contribution of initial conditions to less than 1% of the net hourly $O_3$ concentration over 95% of the available $O_3$ stations.

## 2.4 Evaluation method

We evaluate the simulated concentrations against air quality measurements from the Spanish monitoring stations that are part of the European Environment Information and Observation Network (EIONET; https://www.eionet.europa.eu/). The EIONET network provides a relatively dense geographical coverage of the Spanish territory. During the July 21st-31st episode we used the measurements from 347 stations for $O_3$ and 357 stations for $NO_2$ with a temporal coverage above 85% on an hourly basis. Fig. S2 shows the distribution of the stations for $O_3$ and $NO_2$.

The evaluation based on discrete statistics includes the correlation coefficient (r), Mean Bias (MB), and the Root Mean Square Error (RMSE) (Appendix B). We used the package "openair" (Carslaw and Ropkins, 2012) for R (v3.3.2; R Core Team, 2016) to compute the statistics. We calculate statistics on an hourly basis for $O_3$ and $NO_2$, as well as for the regulatory MDA8 in the case of $O_3$. The evaluation also takes into account the station type, following the categories established by the EEA (i.e., rural background, urban background, industrial and traffic).





There are no direct evaluation methods for apportioned pollutants. Instead, we designed a diagnostic plot for source apportionment analysis at each individual receptor, including a time series of measured and observed $O_3$ and $NO_2$ concentrations together with the simulated tagged sources. In addition, this plot includes the simulated wind speed and direction. These plots are helpful as they compare the modelled $O_3$ and $NO_2$ with the observations, while highlighting the

sources and circulation patterns at least partly responsible for the model behaviour. This work will only discuss in detail the source apportionment plots at key $O_3$ receptor regions, given the high number of stations (260) that simultaneously measure $O_3$ and $NO_2$.

## 3 Results

### 3.1 Description of the ozone episode

In 2012, the largest $O_3$ episode in Europe occurred between July 21[st] and 31[st]. That period alone comprised 33% of the total number of exceedances of the information threshold, and 12% of the number of exceedances of the $O_3$ Target Value in 2012 (EEA, 2013). In Spain, around 60% of the annual exceedances also occurred during this period. (As shown in Querol et al. (2016) July is typically the month with the highest number of $O_3$ exceedances in Spain.) The episode affected the central and north of the IP (Fig. 2c), in particular the 90[th] percentile of the MDA8 $O_3$ concentration exceeded the Target Value at the

surroundings of the Madrid Metropolitan Area (MMA), and an area located north of the Barcelona Metropolitan Area (BMA), which correspond to hotspots and the tail end of large urban and/or industrial plumes.

Figure 2a shows the observed MDA8 $O_3$ concentrations trends at the Spanish EIONET stations during the (extended) summers (i.e., from April to September) from 2000 to 2012, together with the values recorded during the selected episode.

We have categorized the MDA8 $O_3$ concentrations by station type. Rural background stations show the highest MDA8 $O_3$ concentrations during the episode followed by industrial, traffic and urban stations, which is consistent with the observed previous summer patterns. More specifically, the episode shares a similar 75[th] percentile (75p) of the MDA8 $O_3$ concentrations at rural background stations above the Target Value with the particularly severe summer of 2003 (Solberg et al., 2008). Note that the MDA8 $O_3$ concentrations at rural and urban background sites barely changed between 2000 and

2012, which suggests that the benefits from European emission controls may have been significantly counterbalanced by increasing background $O_3$ (Monks et al., 2015). In contrast, the MDA8 $O_3$ concentrations at TR and IN sites slightly increased, likely because of a lower $O_3$ titration due to the preferential abatement of NO vs. $NO_2$. However, we need more research on $O_3$ source apportionment to confirm this hypothesis (Querol et al., 2016; Sicard, et al., 2016).

Figure 2b shows the number of observed exceedances of the $O_3$ Target Value per day during the selected episode. There were more than 100 exceedances in most of the days, with relative maxima on July 25[th], 28[th] and 31[st] attributed to the change in the synoptic conditions. Figure 3 shows the meteorological patterns (2m temperature, 10m wind, precipitation, mean sea





level pressure and geopotential height at 500 hPa) modelled by WRF-ARW during the three distinctive days over the outer EU12 domain.

Stagnant conditions and northwestern advections are the most frequent summer synoptic circulation patterns over the IP, occurring ~44% of the days in a year (Jorba et al., 2004; Valverde et al., 2014). Stagnant conditions are characterized by reduced surface pressure gradients and weak synoptic winds, intense solar radiation, and the development of the Iberian Thermal Low (ITL). The ITL forces the convergence of surface winds from the coastal areas towards the central plateau enhancing sea breezes and mountain-valley winds and subsidence over the Western Mediterranean Basin as described by (Millán et al., 1997, 2000; Millán 2014). In contrast, northwestern advections (NWad) transport air masses from the Atlantic towards the north and west of the IP and they are characterized by atmospheric instability and intense ventilation. Periods of accumulation and venting of pollutants follow the same sequence of pressure ridging and throughing respectively, of the lower and middle troposphere of the IP during the warm season (Querol et al., 2017). The selected episode started with the development of an accumulation- ITL period (July 21$^{st}$-25$^{th}$), followed by a NWad-venting period (July 26$^{th}$-29$^{th}$) and ended with the development of another ITL (July 30$^{th}$-31$^{st}$).

Figure 4 shows the 90$^{th}$ percentile (90p) of the simulated hourly $O_3$ and $NO_2$ concentrations corresponding to the three distinctive days with the relative maxima of exceedances. In the northern Spanish Mediterranean areas, intense $O_3$ episodes often affect the plains and valleys located to the north of the BMA in summer (Toll and Baldasano, 2000; Gonçalves et al., 2009; Valverde et al., 2016a; Querol et al., 2017). High $NO_x$ concentrations from the BMA combined with high summer biogenic VOC levels are driven inland by sea breezes and mountain valley winds, channelled by north-south valleys towards an intra-mountain plains located 60 km north of the BMA. This happened on July 31$^{st}$ when the highest 90p of the hourly $O_3$ concentrations (160-180 µgm$^{-3}$) in Spain occurred over N and NW BMA. Occasionally, as it happened on July 25$^{th}$, anticyclonic winds over the western Mediterranean Sea deflect the sea-breeze flow enriched with precursors from the BMA towards the Gulf of Lion where it reaches the highest 90p of the hourly $O_3$ concentrations (160-180 µgm$^{-3}$) in the IP Mediterranean region. Eastern Spanish Mediterranean areas show similar $O_3$ dynamics, with inland regions depicting the highest $O_3$ peaks (140-160 µgm$^{-3}$) when low-pressure gradients cover central and eastern IP.

In the centre of the IP intense $O_3$ episodes occurred during the development of the ITL, where the affected area depends on the synoptic conditions. Under the absence of synoptic forcing (e.g., July 25$^{th}$), the MMA had the highest 90p of the hourly $O_3$ concentrations (~140-160 µgm$^{-3}$). In contrast, when mountain valley winds are reinforced with synoptic westerlies (e.g., July 31$^{st}$) (Fig. 3) the urban $NO_x$ plume is channelled along the mountain ranges in Madrid towards the northeast and the highest 90p of the hourly $O_3$ concentrations are found along the valley (~140-160 µgm$^{-3}$).



In the northern and northeastern of the IP, the 90p of the hourly $O_3$ concentrations show a significant increase when the blocking anticyclone over Western Europe is combined with the development of the ITL (e.g. July 25th). The stagnant conditions favour the accumulation of $O_3$ precursors around main cities and industrial areas and enhance the local $O_3$ formation.

The NWad pattern (e.g., July 28th) significantly decreases the 90p of the hourly $O_3$ concentrations in the central and northern IP. The northwesterly winds decrease the temperature and therefore the $O_3$ formation. As consequence, $O_3$ levels are reduced in the plumes from the BMA and the MMA, although they are still significant in the latter. Overall, the 90p of the hourly $O_3$ concentrations during the NWadv pattern were ~100 $\mu gm^{-3}$ in most background areas. In contrast, during the ITL it was

above 120 $\mu gm^{-3}$.

## 3.2 Statistical evaluation

CALIOPE has been evaluated in detail elsewhere (Pay et al., 2014 and references therein). Here, we evaluate the updated version of CALIOPE using ISAM to quantify the system's ability to reproduce $O_3$ and $NO_2$ concentrations. Table 2 compiles the quartiles of the statistics calculated by station type. (Note that there are stations not fitting any of the four categories (i.e.,

IN, TR, UB and RB), so the exceedance/station numbers in the IN/TR/UB/RB rows do not sum up to the numbers in the "ALL" rows.)

The model slightly overestimates the hourly and MDA8 $O_3$ concentrations with MB of +12 $\mu gm^{-3}$ and +6 $\mu gm^{-3}$, respectively. The $r$ is above 0.6 in more than 50% of the stations and above 0.7 in 25% of them. The MB for hourly and

MDA8 $O_3$ concentrations are lower at RB stations (±4 $\mu gm^{-3}$) than at IN, TR and UB stations (between +6 and +16 $\mu gm^{-3}$) in 50% of the stations. As expected, the highest number of exceedances of the $O_3$ Target Value was recorded at RB stations (260 exceedances) followed by IN stations (204 exceedances).

At RB stations, hourly $O_3$ is overestimated (+4 $\mu gm^{-3}$) and MDA8 $O_3$ is underestimated (-4 $\mu gm^{-3}$), which indicates that

nighttime $O_3$ is overestimated. The nighttime overestimation is a common feature of CTMs and it is typically attributed to the underestimation of vertical mixing during nighttime stable conditions and to underestimation of the $O_3$ titration by NO (Bessagnet et al., 2016; Sharma et al., 2017).

CALIOPE underestimates the hourly $NO_2$ concentrations with -7 $\mu gm^{-3}$ at TR stations and -2 $\mu gm^{-3}$ at RB stations. This

partly explains the high overestimation of the hourly and MDA8 $O_3$ concentration at TR and UB stations, as well as the systematic overestimation of hourly $O_3$ concentration at nighttime (due to a lack of $O_3$ titration by NO). The hourly $NO_2$ concentration at TR stations feature the highest $r$ (with 25% of stations above 0.6), which proves the reasonably accurate representation of temporal emission in urban areas by the HERMESv2.0 (Guevara et al, 2014; Baldasano et al., 2011). In



contrast, the RMSE is highest at TR stations, which results from the underestimation of $NO_2$ peaks during traffic rush hours. Underestimation of $NO_2$ traffic peaks is a common problem in Eulerian mesoscale models (Pay et al., 2014), as emission heterogeneity is lost in the grid cell-averaging process, which is especially critical in urban areas. Next generation microscale models will potentially solve this problem (Lateb et al., 2016). Besides the dilution of the emission in the grid, meteorology

also may play an important role in the low performance of $NO_2$ and $O_3$ in hotspot areas. Several inter-comparison studies (e.g., EURODELTA and AQMEII) agree on the limitations of models to simulate meteorological variables that affect the hourly $NO_2$ temporal variability, which controls model performance for $O_3$ in high $NO_x$ environments and areas downwind (Bessagnet et al., 2016; Solazzo et al., 2017).

Figure 5 classifies the hourly and MDA8 $O_3$ concentrations at the air quality stations into four MB categories that account for the 93% of stations. There are two categories with the lowest hourly $O_3$ MB ($\pm 10$ $\mu gm^{-3}$): type A with a MDA8 $O_3$ MB between -40 and -10 $\mu gm^{-3}$ and type B with MDA8 $O_3$ MB of $\pm 10$ $\mu gm^{-3}$. There are two categories with the highest hourly $O_3$ MB (between 10 and 40 $\mu gm^{-3}$): type C with a MDA8 $O_3$ MB of $\pm 10$ $\mu gm^{-3}$ and the type D with MDA8 $O_3$ MB between 10 and 40 $\mu gm^{-3}$. In addition, Fig. S2 depicts the MB, RMSE and $r$ for hourly and MDA8 $O_3$, and hourly $NO_2$

concentrations. The best performances for $O_3$ (type B) are found at the 28% of the stations, located in the surroundings of the MMA, the BMA and most of the northern Mediterranean stations, which is consistent with the highest $r$ ($0.6 < r < 0.9$) found in the center and north of the IP (Fig S3). The highest $O_3$ overestimations (type D) are present at 36% of the stations, mainly located in highly industrialized areas in Spain (Guadalquivir Basin, Strait of Gibraltar, Valencia) and inside the MMA. Next sections analyze the origin of these $O_3$ biases using the source apportionment time series.

The comparison with previous CALIOPE studies (Baldasano et al., 2011; Pay et al., 2014) indicates that $r$ is in the same range for $O_3$ (0.6-0.7) and $NO_2$ (0.4-0.5) at individual stations; the same applies to RMSE (15-29 $\mu gO_3$ $m^{-3}$ and 10-20 $\mu gNO_2$ $m^{-3}$). Modelled $O_3$ shows higher performance at traffic stations in large cities, since stations influenced by road transport emissions (i.e., high-$NO_x$ environments) are better characterized with a more pronounced daily variability (Baldasano et al.,

2011).

### 3.3 Source-sector ozone contributions during peak episodes

Figure 6 shows the 90p of the hourly $O_3$ concentration over the IP tagged by source type (Table 1) for different days (July 25$^{th}$, 28$^{th}$ and 31$^{st}$). (Fig. S3 in the supplementary material shows similar plots for $NO_2$.) The imported $O_3$ is by far the largest contributor showing a 90p ranging from 70 to 120 $\mu gm^{-3}$ in the east/north/centre of the IP on July 25$^{th}$ /28$^{th}$ /31$^{st}$ ,

respectively (Fig. 6). The imported  $O_3$ enters the study domain through the IP4 domain boundaries and it can only be transported, scavenged deposited or depleted by $O_3$ precursors. Therefore, areas with low imported $O_3$ concentrations (< 50 $\mu gm^{-3}$) are good indicators of (1) the accumulation of specific $O_3$ precursors that deplete imported $O_3$, and (2) the subsequent $O_3$ photochemical production that occur mostly under stagnant conditions and around the largest industrial/urban areas. The



90p of the hourly imported $O_3$ concentration shows the lowest values in two different conditions and regions. First, on July 25th in the northwestern IP and Portugal, extremely low winds allow the accumulation of local pollutants that titrate imported $O_3$ concentrations down to 30-70 $\mu gm^{-3}$; at the same time $O_3$ is locally produced due to traffic emission downwind of major northern cities (La Coruña, Gijón, Bilbao) (60-120 $\mu gm^{-3}$), shipping activities (up to 40 $\mu gm^{-3}$) and the generation of energy and industrial processes (10-20 $\mu gm^{-3}$). Second, on July 31st in the northeastern IP, when the pollutants transported from the Gulf of Lion and Catalonia towards the Mediterranean act as a sink of imported $O_3$ reducing its concentration down to 60 $\mu gm^{-3}$. As a result, there is a local $O_3$ formation up to 120-160 $\mu gm^{-3}$ along the Ebro Valley and the Lleida Plain.

After the imported $O_3$, the largest contributor to $O_3$ is the road transport sector. Downwind of the major urban areas in Spain (i.e., Madrid, Barcelona, Bilbao, Sevilla, Valencia, Gijon, Pontevedra), on-road traffic contributed to the 90p of the hourly $O_3$ concentrations as much as 60-120 $\mu gm^{-3}$, and affected different areas depending on the synoptic/mesoscale regimes (Fig. 6). In the northern BMA, the 90p of the hourly $O_3$ concentration from the road transport sector reaches its maximum when the stagnant conditions affects the centre and eastern IP (e.g., July 31st). As noted above, mesoscale winds carry traffic $O_3$ precursors from the BMA inland, channelled by north-south valleys towards the intra-mountain plain in the north. Over the MMA, the 90p of the hourly $O_3$ concentration from the road transport sector showed the maximum when the ITL was combined with the synoptic westerlies (e.g., July 31st), carrying high $O_3$ as far as to the Ebro Valley, as shown in Fig 4.

Regarding the contribution from the non-road transport sector, the Atlantic regions of the IP show the highest 90p of the hourly $O_3$ concentration (25-40 $\mu gm^{-3}$) on July 25th. The stagnant condition favoured the accumulation of precursors from the Atlantic shipping route and the formation of $O_3$ within the region. The Spanish Mediterranean region shows the highest 90p of the hourly $O_3$ concentrations from the non-road transport sector in front of the southeastern coasts of the IP (~180 $\mu gm^{-3}$) when the westerlies in the Strait of Gibraltar injected precursors from international shipping into the Mediterranean Basin (e.g., July 28th and 31st). Note that during days with high 90p of the hourly $O_3$ concentration from non-road transport, the imported $O_3$ concentration shows the lowest 90p due to the NO titration effect over emission areas.

The elevated point source emission sectors (i.e., energy and industry) contributed less to $O_3$ than the traffic sector, but their contributions were significant reaching 15-25 $\mu gm^{-3}$ of the 90p of the hourly $O_3$ concentrations (Fig. 6). The north and northeastern IP, the Mediterranean coast, and the Guadalquivir Valley are the most affected regions under stagnant conditions.

The contribution of the remaining sectors (OTHR) to the 90p of the hourly $O_3$ concentrations was similar to that of the elevated point sources (15-25 $\mu gm^{-3}$), but it reached up to 30 $\mu gm^{-3}$ in areas downwind of Oporto and Lisbon (Fig. 6). OTHR includes the formation of $O_3$ from the remaining anthropogenic and biogenic sources (accounting for less than 8% of total $NO_x$ emissions, but 93% of total VOC). The high OTHR concentration around the biggest cities in Portugal may be related



to precursors emitted by the residential sector (SNAP2 and 9) and biogenic emissions, as found in other source apportionment studies over Portugal (Borrego et al., 2016; Karamchandani et al., 2017).

## 3.4 Regionalization of source-sector contributions

The analysis of the source-sector contributions to $O_3$ allowed identifying ten receptor regions with similar characteristics in terms of meteorological and $O_3$ patterns, main source contributors and geographical patterns (Figs. 4 and 6). Figure 7c shows the location of the air quality stations belonging to each receptor region corresponding to the central IP (CIP), the eastern IP (EIP), the Ebro Valley (EV), the Guadalquivir Valley (GV), the Mediterranean Sea (MED), the northeastern IP (NEIP), the northern IP (NIP), the northwestern IP (NWIP), the southern IP (SIP) and the western IP (WIP). The $O_3$ receptor regions are consistent with Diéguez et al. (2014) and Querol et al. (2016), who proposed a similar regionalization based on observations from air quality stations.

Figure 7a shows the absolute $O_3$ contribution of each tagged source at air quality stations by region along with the modelled and observed daily mean concentration during exceedances of 120 μgm$^{-3}$ of the observed MDA8 ozone; Fig. 7b shows the respective fractional mean $O_3$ contribution. (Table S2 compiles de numerical values of Fig. 7.) We have excluded the ICON $O_3$ in Table S2 because its contribution is negligible after six days of spin-up. The spatial $r$ between modelled and observed daily mean $O_3$ concentration during exceedances of the MDA8 target value is 0.79, which indicates that CALIOPE is able to reproduce reasonably well the spatial variability of $O_3$ during peak episodes in Spain.

The MED region represented by stations in the Balearic Islands shows the highest imported $O_3$ contribution (76%) because it is relatively far away from important anthropogenic $NO_x$+VOC sources in the IP. Under the ITL influence (July 25$^{th}$ and 31$^{st}$), MED received air masses enriched with on-road traffic precursors from southern France and the NEIP, which enhanced $O_3$ production up to 7%. Shipping emissions in the Mediterranean contributed up to 8% of the $O_3$.

After MED, there is a cluster of regions along the Spanish Mediterranean coast (i.e., NEIP, EV and EIP) showing imported $O_3$ contributions between 60 and 68 % of the daily mean $O_3$ under exceedances. This is explained by their proximity to the eastern boundary and the frequent mesoscale phenomena enhancing the recirculation and accumulation of imported $O_3$ along the Spanish Mediterranean coasts. The contribution of road and non-road transport is similar (~11-16%) because these regions have both important roads and maritime trade routes. Note that the SIP region, which is also located in the Spanish Mediterranean coast, shows a daily mean imported $O_3$ concentration lower than the other Mediterranean regions (~57%) and the highest non-road transport contribution in Spain (19%). The main sink of imported $O_3$ are precursors resulting from dense shipping traffic through the Strait of Gibraltar, which have substantial impact in the $O_3$ production downwind (either in the Alboran Sea or the Gulf of Cadiz).





The regions including the largest metropolitan areas in Spain are the CIP (Madrid) and the NEIP (Barcelona). Both regions show an imported $O_3$ contribution of ~60% and a similar contribution from the road transport sector (18 and 16%, respectively). However, the NEIP shows a slightly higher contribution from non-road transport (13 vs 10%) due to the influence of international shipping near coastal areas.

The northern and northwestern regions (NIP and NWIP) had relatively lower imported $O_3$ contributions (56-59%). The contribution of non-road transport was ~10-12%, slightly lower than in the Mediterranean coast, and that of road transport was also significant (~14-15%). The contribution from the industrial sector was one of the highest in the country (~5%) due to the influence of the large industrial facilities located in several areas of Galicia, Cantabria, Asturias, and the Basque

Country. The contribution from the energy sector in the NWIP region was the highest in Spain (~5%) due to emissions from large coal-fired power plants, including As Pontes, Meirama, Aboño, Guardo, and Compostilla.

The WIP had the lowest daily mean $O_3$ concentrations during days exceeding the $O_3$ Target Value (93.5 µgm$^{-3}$) and a high imported $O_3$ contribution (~63%). $NO_x$ emissions in the WIP region are moderate (Fig. 4), which could explain the low daily

mean $O_3$ concentration. There is a significant contribution from traffic (14% for road transport and 11% for non-road transport) and industrial and energetic sectors (7%) to the daily mean $O_3$ concentrations. These anthropogenic contributions suggest that $O_3$ in the WIP is produced by precursors transported from the surrounding cities (Oporto, Lisbon and Madrid) and the highly industrialized areas in the NWIP and the NIP (Fig. 6).

The Guadalquivir Valley had the lowest imported $O_3$ contribution of the IP (~46%) and the highest daily $O_3$ concentration during days of exceedance. The relatively low imported $O_3$ contribution resulted from the titration effect driven by the high $NO_x$ emissions from industrial and urban hotspots in the region. The on-road traffic was the highest anthropogenic contributor to $O_3$ (~18%) due to the emissions from three major cities (Sevilla, Huelva, and Cordoba). Although $O_3$ in Huelva may be overestimated (as discussed later), shipping is the second most important contributor to $O_3$ in the

Guadalquivir Valley (~17%) due to the important fluvial transport along the Guadalquivir River. (The Guadalquivir River is one of the most important routes for merchandise transport in Europe.) In fact, the non-road transport sector is the highest contributor (~17-19%) in Southern Spain, both in the Guadalquivir Valley and the SIP, also due to the dense maritime routes across the Strait of Gibraltar.

The following sections analyse the source apportionment results eight stations located in key regions (see Fig. 5). We analyse two regions with high on-road traffic contribution (CIP and NEIP), and two regions with high contribution from industry and energy production (NWIP and Guadalquivir Valley).



### 3.4.1 Central Iberian Peninsula

Figure 8 show the source apportionment time series of the hourly $O_3$ and $NO_2$ concentrations at two stations in the CIP, an urban station in Madrid (station 1 in Fig. 8a), and another one located in Guadalajara (station 2 in Fig. 8a), which is a medium size city affected by Madrid's urban plume. The model reproduces reasonably well the main processes and source

contributions, with a significant imported $O_3$ contribution in both stations.

In Madrid, the model reproduces the $O_3$ traffic cycle featuring the typical low $O_3$ concentrations ($< 40$ µgm$^{-3}$) in the early morning and in the afternoon due to $O_3$ titration (Fig. 8a). However, $O_3$ was overestimated (MB type D) during daytime peaks due to the overestimation of the $NO_2$ morning peaks during stagnant conditions. The $NO_2$ overestimation correlates

with the highest road transport contribution. The $NO_2$ overestimation in Madrid is inconsistent with the overall underestimation of $NO_2$ in urban areas due to the dilution effect. The results point towards a poor representation of the vertical mixing during the stagnant conditions. In urban areas downwind (Fig. 8b) the modelled $O_3$ shows a positive bias (MB type B) due to the underestimation of $NO_2$. While traffic $NO_2$ emissions in the largest urban areas (i.e. Madrid, Barcelona) are estimated at the road link level using specific data such as daily average traffic, mean speed circulation and

real-world vehicle fleet composition profiles, in medium size cities this information is not available and traffic emissions are estimated using emission factors that depend on population density (Guevara et al., 2013), which implies having a larger uncertainty in the results.

The $O_3$ contribution from the industrial sector (whose precursors could come from facilities in the southern MMA, Fig. 4)

reinforces the $O_3$ peaks up to ~10 µgm$^{-3}$, meanwhile the contribution from non-road transport increases systematically the background $O_3$ concentration by ~15 µgm$^{-3}$. The $O_3$ contribution from non-road transport in this region may arise from the Atlantic shipping route, Madrid's airports and the agricultural machinery operating in the surrounding rural areas.

Figure 8a and 8b also shows that the urban $O_3$ plume presents different intensity and distribution depending on the

meteorological pattern. Downwind of Madrid (Fig. 8b), stagnant conditions favour $O_3$ peaks exceeding 120 µgm$^{-3}$ and on-road transport contributes up to 20-30 µgm$^{-3}$ (e.g., July 25$^{th}$). The highest $O_3$ concentrations (~160 µgm$^{-3}$) are modelled when westerly winds channelled along the Tajo valley carry the polluted air masses in a NE direction through the Tajo Valley, which results in a $O_3$ contribution of ~70 µgm$^{-3}$ from road transport sector in areas downwind (see wind vectors in Fig. 8b on July 28$^{th}$ and 31$^{st}$).

### 3.4.2 North-Eastern Iberian Peninsula

Figure 8c and 8d show the source apportionment time series at two stations in the NEIP, an urban station in Barcelona (station 3) and a remote rural area downwind (station 4). In the urban station, $NO_2$ levels of up to 100 µgm$^{-3}$ affect $O_3$



concentrations by titration during traffic peaks. In contrast, the rural station downwind depicts a higher $O_3$ and lower $NO_2$ concentration than the urban station.

Absolute $O_3$ biases in both stations are ~10 µgm⁻³ (MB type B). In the urban area, the modelled $O_3$ concentrations are in
agreement with the observations. The model suggests that $O_3$ mostly results from import and from the NO titration effect due to local road transport and industrial sources (Fig. 8c). However, the model presents some under/overestimation at nighttime, which could result from the $NO_2$ underestimation and complex mesoscale phenomena not captured by the model. Note that the $O_3$ diurnal cycle in the NEIP urban areas is less marked than in the CIP due to persistently high $O_3$ concentration at night (~60 µgm⁻³). The breezes and mountain-valley winds contribute to the accumulation and recirculation of pollutants in this
region.

Despite of the slight overestimation of $NO_2$ concentration in rural areas, modelled $O_3$ peaks (> 120 µgm⁻³) are highly correlated with observations (Fig. 8d), which suggests that overall the model reproduces the main transport paths, photochemical processes, and relative contributions from different sources. Imported $O_3$ is one of the main contributors to
ground-level $O_3$ (from 40 to 100 µgm⁻³), but during peaks the on-road traffic contribution sharply increases up to 80 µgm⁻³.

The $O_3$ concentration from the road transport sector arriving at rural areas in the NEIP, mainly comes from the Barcelona and surroundings as a result of the afternoon sea breezes (see wind vectors in Fig. 8c and 8d). However, under specific meteorological patterns these winds also carry precursors from other cities located in the NW Mediterranean Basin. For
example, on July 28th both the model and observations show that $O_3$ exceeded 120 µgm⁻³ in both urban and rural areas and the model attributed ~60% of it to regional anthropogenic sources (non-imported $O_3$). In the morning, two main meteorological phenomena favoured the transport of precursors towards the Western Mediterranean. First, the North African Thermal Low reinforced the sea breezes in the NEIP coast. Second, NW synoptic winds channelled between the Pyrenees and the French Central Massif towards the Golf of Lion transported precursors from urban areas upwind (e.g., Marseille,
Toulouse). In the afternoon, the sea breezes transported air masses enriched with $O_3$ and precursors inland. Other authors (Gangoiti et al., 2001) have hypothesized that high $O_3$ concentration in the Western Mediterranean Basin is influenced by transport from France via the Carcassonne gap. The present experiment cannot quantify the contribution of French cities to the $O_3$ concentration over the NEIP, but future studies could explicitly tag the emission from the French regions.

### 3.4.3 Guadalquivir Valley

The largest modelled $O_3$ overestimations are found in the Guadalquivir Valley (MB type D). Precursors emitted along the Guadalquivir Valley arise from large industrial facilities and a coastal power plant (Fig. S3), a few densely populated cities (notably Sevilla and Cordoba with more than 700,000 and 320,000 inhabitants, respectively), and the navigable river. The airflow in the region is controlled by the Atlantic synoptic conditions, the development of breezes in the coast of the Gulf of



Cadiz and the channelling through the Guadalquivir Valley (Adame et al., 2008). We have selected two stations along the Guadalquivir Valley to evaluate $O_3$, one in the urban area of Sevilla (station 5 in Fig. 9a), and one in a rural coastal area (station 6 in Fig. 9b). The contribution of anthropogenic sources to $NO_2$ at the coastal rural background station is low but still significant. The contribution of non-road transport is due the influence of one of the largest Spanish harbours. The contribution of the energy sector to the $O_3$ concentration is also noticed (e.g., July 25th). As expected, the urban station (Fig. 9a) shows a high $NO_2$ concentration dominated by on-road traffic. $NO_x$ from traffic is the main sink of $O_3$ in the city and the model reproduces the titration effect in agreement with observations (e.g., July 28th-31st).

High $O_3$ overestimations (10-30 $\mu gm^{-3}$) at both stations (Fig. 9a and b) were detected during July 25th-28th which corresponds to intense and persistent SW winds transporting air masses from the Atlantic Sea along the Guadalquivir Valley, as shown by the wind vectors in Fig. 9a and b. Although the model overestimates $O_3$ concentrations, it reproduces the temporal variability. Our results suggest that the non-road transport sector is a significant contributor along the Guadalquivir Valley during these days. The impact of shipping emission on $O_3$ in the Guadalquivir Valley region is mainly evidenced by the relative high $NO_x$ from ship exhaust (Fig. S3). $NO_2$ time series at the coastal station (Fig. 9b) indicates that the model overestimates $NO_2$ concentrations in hours where the $NO_2$ contribution from non-road transport is the highest. The unrealistic $NO_2$ peaks from non-road transport suggest that shipping emissions are overestimated in the HERMESv2.0 model. HERMESv2.0 uses the EMEP gridded emission database at 50 km x 50 km horizontal resolution to estimate shipping emissions and spatially distributes them to the 4-km model domain using the marine routes reported by Wang et al. (2008). Marine traffic emissions estimated with current state-of-the-art methods, such as the use of data provided by Automatic Identification System (AIS), show large discrepancies with the reported CEIP-EMEP gridded inventories both in term of totals and spatial distribution (Jalkanen et al., 2012), suggesting that uncertainties associated to current official emissions may be large. This factor, added to the 15% decrease of $NO_x$ shipping emissions observed in Europe between 2009 (HERMESv2.0 base year) and 2012 (http://www.ceip.at) could explain the discrepancies observed.

$O_3$ peaks are highly biased (by ~40-50 $\mu gm^{-3}$) during stagnant conditions on July 28th-31st, especially in the urban station (station 5). During these days, measured $NO_2$ daily peaks increased up to 40 $\mu gm^{-3}$. Although the model reproduces the variability, it overestimates the concentration due to an excessive on-road transport contribution. As discussed for the CIP, under low dispersion conditions $NO_2$ concentrations are difficult to model due to complex mixing regimes.

### 3.4.4 Northwestern Iberian Peninsula

Figures 9c and 9d show the source apportionment time series at one urban (station 7) and one rural (station 8) background station in the NWIP. The urban station (Fig. 9c) located in Santiago de Compostela, a medium size city with ~100,000 inhabitants, shows a high $NO_2$ concentration with a dominant contribution from the road transport sector. Traffic $NO_2$ is the



main sink of urban $O_3$ via titration. Because $NO_2$ is underestimated, especially during stagnant conditions (July 24[th]-27[th]), $O_3$ concentrations are overestimated (MB category C).

In rural background areas (Fig. 9d), the modelled $O_3$ concentration is in general agreement with observations (MB category B), but the model tends to underestimate the observed concentrations, particularly under weak winds. $NO_2$ is likely to be underestimated due to missing traffic emissions. As noted previously, traffic emissions are poorly constrained in small and medium size cities, due to a lack of detailed information. There is also additional uncertainty in the precursors emitted from the large coal power plants and industries in the region (Valverde et al., 2016b). Our study uses emissions for 2009, and it has been estimated that between 2009 and 2012 energy production in coal-fired power plants increased from 13.1% to 19.4% (UNESA, 2012).

Despite the $O_3$ biases during stagnant conditions, the time series show that model reproduces reasonably well the observed $O_3$ variability under different synoptic conditions. $O_3$ reaches the highest concentration (~100/150 $\mu gm^{-3}$ in urban/rural areas) under stagnant conditions (July 24[th]-27[th]) when the contribution of anthropogenic sources from all activity sectors is the highest (60-70%). $O_3$ concentrations decrease down to ~70 $\mu gm^{-3}$ under NW advective conditions (e.g., July 28[th]-30[th]) when the imported $O_3$ shows the highest contribution (80-90%). Saavedra et al. (2012) found that stationary anticyclones over the northern play an important role in the occurrence of high $O_3$ concentrations over the NWIP. Our results show that under these stagnant conditions $O_3$ concentrations are due largely to in situ production (photochemistry) from on-road traffic, shipping, power plants, and industry in almost the same proportion.

### 3.5 Imported ozone

These $O_3$ source apportionment results indicate that imported $O_3$ represents the highest contribution to ground-level $O_3$ concentration in southwestern Europe. Imported $O_3$ enters the IP4 domain through the boundaries; it includes the contribution of $O_3$ from the EU12 domain, which in turn includes the contribution of hemispheric $O_3$ from the MOZART-4 global model. The imported $O_3$ is as large as the background $O_3$ regionally produced within the IP. We note that the small biases at rural background stations obtained in the evaluation section indicate an overall high performance of the modelled background $O_3$ in the IP4 domain (Fig. S1). Given its importance, this section is devoted to analyse the importance of the imported $O_3$ in southwestern Europe. In particular, we aim to understand its high contribution within the IP, far from the model domain boundaries.

Figure 10 shows the vertical cross-sections at 6, 12, and 18 UTC for $O_3$ and $NO_2$ at a constant latitude (40.38° N) on July 25[th], 28[th] and 30[th]. The model predicts a pronounced $O_3$ vertical gradient above a height of 4 km above sea level (asl), showing that $O_3$ in the free troposphere is, to a large extent, imported to the IP4 domain (Safieddine et al., 2014). As





expected, $NO_2$ mixing ratios show a negative gradient with altitude as it is mainly emitted at the surface. In the morning, the sun starts to heat up the ground, producing convective thermals and forcing the growth of the mixing layer. At noon, the mixing height reaches its maximum, being the highest in the CIP (2-4 km) and decreasing towards the coast (< 1 km) (Fig. 10). At the mixing layer top the $O_3$-enriched air aloft is entrained into the mixing layer, mixing with $O_3$ and other pollutants

produced locally within the mixing layer. When the mixing height decreases, $O_3$ is left in the free troposphere forming high $O_3$ residual layers (Gangoiti et al., 2001) that contribute to the regional transport. Over the following days, these residual layers (composed of imported $O_3$ and local $O_3$ produced within the domain over previous days) can be entrained by fumigation into the mixing layer to reach the surface. This fumigation effect, previously described in the eastern USA (Zhang and Rao, 1999; Langford et al., 2015) and in the Western Mediterranean Basin (Kalabokas et al., 2017; Querol et al.,

2018), leads to a rapid increase in $O_3$ concentrations at ground level. The accumulation and recirculation of air masses is intensified along the Eastern Mediterranean coast (Millán et al., 1996, 2000; Gangoiti et al., 2001; Querol et al., 2017) by the action of the breezes and mountain-valley winds. Furthermore, the small deposition velocity of $O_3$ over the sea and its high atmospheric lifetime in the free troposphere contributes to enrich the $O_3$ background concentration (c.a. several weeks, Monks et al., 2015; Seinfeld and Pandis, 2016).

The $O_3$ fumigation effect was particularly intense on July 30th-31st, when a high $O_3$ levels are found in the free troposphere compared to previous days (Fig. 10). The analysis of the $O_3$ concentration map with the imported $O_3$ contributions (Fig. 6) indicates that ground-based $O_3$ is neither advected nor titrated; therefore, it can only result from vertical mixing. The high $O_3$ mixing ratio in the free troposphere was mainly due to $O_3$ advection entering the Atlantic boundary driven by westerlies (Fig. 3). We hypothesize that two events may have contributed to the increase of the $O_3$ concentration by long-range transport.

First, a low-pressure system in the British Islands on July 28th could have transported significant $O_3$ amounts from the stratosphere to the free troposphere (Fig. S4). Second, $O_3$ episodes generated in mid-July over the Eastern USA predicted by the MOZART-4 model could have contributed to an increase in the $O_3$ transported from N America to Europe by the action of the prevailing westerlies (Fig. S4) associated with cyclonic systems along the ''warm conveyor belt'' (Pausata, et al.,

2012; Derwent et al., 2015).

**4 Discussion and conclusions**

We investigate the origin of ground-level $O_3$ concentration over southwestern Europe during the episode July 21st -31st, 2012. The period includes the two most frequent summer synoptic circulation patterns over the region: stagnant conditions with development of the Iberian Thermal Low (ITL) and North Western Advection (NWad). We used the Integrated Source

Apportionment Method (ISAM) within the CALIOPE system to estimate the main sources responsible of the high $O_3$ concentrations in receptor regions over the IP compared to the imported (regional and hemispheric) $O_3$ contribution. In addition, the source apportionment method has allowed an in-depth evaluation of the modelling system applied. Below, we




summarize the new findings from the source attribution study, we discuss the model uncertainties and we give a perspective on some of the regulatory implications, which may be applicable to other non-attainment regions.

### 4.1 Source apportionment results

The results demonstrate that the $O_3$ problem over the Western Mediterranean Basin is local, regional, and hemispheric.
Long-range transport of $O_3$ from beyond the IP domain is the main contributor to the ground-level daily mean $O_3$ concentration (~45%) during peak episodes. The imported $O_3$ contribution ranges from 40% during $O_3$ peaks to 80% at night or during well-ventilated conditions. The absolute imported $O_3$ is higher in NEIP than in the central IP due to the recirculation and accumulation of pollutants along the Mediterranean. The high imported $O_3$ at the surface far away from the model boundaries is consistent with the high levels of $O_3$ in the free troposphere (resulting from local/regional layering and accumulation, and continental/hemispheric transport) along with intense vertical mixing during the day.

During high $O_3$ events, the imported $O_3$ is added to the local and regional anthropogenic contributions. The road transport is an important contributor to the $O_3$ concentration in rural areas downwind of the large cities in Spain. It contributed up to 16-18% of the daily mean $O_3$ concentration under exceedances of the target value for human health protection, and up to 70 $\mu gm^{-3}$ on an hourly basis downwind of Barcelona and Madrid. The non-road transport sector is as significant as road transport inland (10-19% of the daily mean $O_3$ concentration during the peaks). There is a high influence of international shipping (13%), affecting the coastal areas in the Mediterranean and the southern IP (along the Strait of Gibraltar) with contributions of up to ~20 and ~30 $\mu g/m^3$, respectively. The energy and industrial sectors contribute ~6-11% of the daily mean $O_3$ concentration during the peaks and over all the receptor regions. As they usually are injected in high altitudes, their contribution extend way beyond their surroundings. The energy combustion sector (3%) and industrial and non-industrial combustion sectors (3%) have a mean contribution of 2-4 $\mu gm^{-3}$, going up to 4-6 $\mu gm^{-3}$ in $NO_x$-limited areas (i.e., the western IP). The OTHR tag, which lumped the remaining sectors (i.e., SNAP 2, 5, 6, 9, and 11, see Table 1), is the fourth main contribution to the daily mean $O_3$ concentration during the days exceeding the target value (~2-8%). Future work should tag the biogenic sources as an individual sector as it is the main contributor to VOC emissions in Spain (i.e., ~70% in 2009 according to HERMESv2.0 model).

### 4.2 Model uncertainty

The modelled concentrations present uncertainties that are in the same range as the most recent inter-comparison studies using state-of-the-art air quality models. Our results show a good agreement between the modelled and observed $O_3$, which provides confidence in our source apportionment results.

In addition, our model evaluation together with the source apportionment results has allowed a better understanding of the origin of model errors. Under stagnant conditions, the model shows lower performances: in coastal areas, this may be related



to a poor representation of mesoscales processes such a sea breezes and recirculation; in Madrid, $NO_2$ overestimations may be due to a poor representation of the vertical mixing. There is likely an overestimation of shipping emissions, and an underestimation of road transport emissions in medium size cities.

Overall, the estimated imported $O_3$ depends on both the performance of the CALIOPE system over EU12, and on MOZART-4 at global scale. The evaluation of model in the EU12 domain using the European EIONET stations (not shown here) showed good skills for $O_3$ and its precursors. Furthermore, the source apportionment results have shown that $O_3$ local production could be as large as the imported $O_3$, so the small biases at rural background stations indicate an overall high performance baseline background $O_3$ in the IP4 domain (Fig. S1). This study has not evaluated MOZART-4, but there are

many studies supporting its good performance. Several inter-comparison studies (e.g., EURODELTA and AQMEII) also found the importance of chemical boundary conditions to reproduce the $O_3$ background concentrations (Giordano et al. 2015; Bessagnet et al., 2016; Solazzo et al., 2017).

Our set-up involves some uncertainty in the estimation of the imported contribution of $O_3$. In reality, there is a fraction of the

imported $O_3$ that may have been generated within the IP4 domain before the period of simulation (including the spin-up). We have assumed that this fraction is negligible and future works should check to what extent this assumption is correct.

### 4.3 Regulatory implications

Our study has identified the main sources responsible of high $O_3$ in receptor regions in the Western Mediterranean Basin, but cannot predict whether emission abatement will have either a positive or negative effect in $O_3$ changes due to the non-

linearity of the $O_3$ generation process (Monks et al., 2015 and papers therein). Subsequent source sensitivity analysis tailoring the identified main contribution sources could predict how $O_3$ will respond to reductions in precursor emissions. Future source sensitivity studies are essential to define the most efficient abatement strategies to reduce high $O_3$ concentrations in the Western Mediterranean Basin. Complementarily, it would be important to know where the emission abatement strategies should be applied. To answer this question further source apportionment studies should be conducted

tagging both source activities and source regions.

This case study supports a relatively old conclusion by the European Commission (EC, 2004) pointing out that the effectiveness of abatement strategies for achieving compliance with the European air quality standards in southern Europe might be compromised by the long-range transport of $O_3$. This is especially true in Mediterranean regions (i.e., NEIP, EV

and EIP) where the contribution of imported $O_3$ is particularly dominant (60-68% of the daily mean $O_3$ concentration in episodes) as a result of the accumulation and recirculation of pollutants over the Mediterranean Basin. In those areas, if the long-range transport of $O_3$ is not reduced, the mean background level will not decrease making it more vulnerable to exceedances of the $O_3$ target values by enhanced local production under stagnant conditions.





Reduction of the background $O_3$ concentration can be achieved by decreasing the concentrations of $O_3$ and precursors aloft. Given the polluted dome aloft the Western Mediterranean Basin during a typical $O_3$ episode, which may result from local/regional layering and accumulation, and continental/hemispheric transport, it will necessary and important to

implement emission reduction strategies on the hemispheric and continental scale. There is an urgent need of international collaboration at European and hemispheric level to reduce $O_3$ and its precursors. High background contribution to Iberian $O_3$ could represent a substantial future challenge to the attainment of $O_3$ limit values, which requires controlling hemispheric $O_3$ (EC, 2004; Fowler et al., 2013).

Besides the long-range transport contribution, $O_3$ peaks during episodes result from the local/regional photochemical formation from the transport and industrial sectors, and therefore regional emission reduction strategies are also urgent. The contribution from road transport determines $O_3$ peaks in rural areas (up to 70 µg/m$^3$) downwind of the main Spanish Metropolitan Areas (i.e., Madrid and Barcelona). To reduce $O_3$ peaks downwind of the Madrid city, a decrease of road transport emissions in the urban area would be effective. However, to reduce $O_3$ peaks downwind of the Barcelona city, the

reduction of road traffic emission in the city should be complemented with abatement strategies in other Mediterranean cities, especially in Southern France.

In the BMA the contribution from energy and industrial sectors to $NO_2$ concentration can be in the same range than the contribution from road transport (~40-60%, Fig. 8c). In contrast, in areas downwind of the BMA the contribution from

energy and industrial sectors to $O_3$ concentrations is relatively low compared with the contribution from road transport (Fig. 8d). The differences contribution to O3 concentration might link with the different reactivity of VOC for $O_3$ formation. Each VOC emission source emits a different mix of VOC, which makes a different contribution to photochemical ozone formation. For example, in the UK, Derwent et al. (2007) showed a higher photochemical $O_3$ creation potential for road transport emissions than for production processes and combustion. Future national policy actions to control the emissions of

VOC should tackle the sources that contribute more to photochemical $O_3$ formation.

Although the non-road transport contribution was found to be overestimated in coastal areas in the south of the IP in the present experiment, it cannot be neglected and actions controlling international shipping should be considered as important as those related with road transport especially in regions with big harbours (e.g., Huelva and Barcelona). Dalsøren et al.

(2010) indicated that annual $O_3$ concentration is increasing yearly in a range of 0.5-2.5 ppb in areas impacted by shipping activities. Recent studies indicate that shipping emissions are projected to increase significantly due to increases in transportation demand and traffic. As the Strait of Gibraltar is the only shipping route connecting the Atlantic Ocean with the Mediterranean Basin, the regulation of these emissions is key in order to control $O_3$ exceedances in Spain and the Mediterranean Basin. Shipping emissions can be regulated by each country within 400 km of coastlines, but policy-induced



controls for offshore emissions are very dependent on the success of adopted and proposed regulations within the International Maritime Organization.

In highly industrialized regions (i.e., Guadalquivir Basin and northwestern IP), abatement strategies affecting all sectors at regional scale could contribute to decrease the local formation of $O_3$ as the regional/local anthropogenic contribution can be greater than 50% during several days.

Overall, we find that the imported $O_3$ is the largest input to the ground-level $O_3$ concentration in the IP during episodes. In addition, during stagnant conditions, the emission from local anthropogenic activities in the IP control the $O_3$ peaks in areas downwind of the main urban and industrial regions. Furthermore, ground-level $O_3$ concentrations are strongly affected by vertical downward mixing of $O_3$-rich layers in the free troposphere, which result from local/regional layering and accumulation, and continental/hemispheric transport. The importance of both imported and local contributions to the $O_3$ peaks in the IP demonstrates the need for detailed quantification of both contributions to high $O_3$ concentrations for local $O_3$ management. Furthermore, the influence of local sources and topographical and meteorological conditions in the high $O_3$ concentration indicate the importance of designing $O_3$ abatement policies at local scale.

This work has identified local and imported contributions to $O_3$ concentration during an episode in a particular area in southwestern Europe. However, further detailed and longer $O_3$ source apportionment studies targeting other nonattainment regions in Europe are necessary prior to define local mitigation measurements that complement national and European-wide strategies to reach the European $O_3$ objectives.

**Data availability**

Air quality measurements are available at the EIONET database (https://www.eea.europa.eu/data-and-maps/data/airbase-the-european-air-quality-database-7). The CMAQ code is available at the https://www.cmascenter.org/. The WRF-ARW code is available at http://www2.mmm.ucar.edu/wrf/users/download/get_source.html. Download Mozart-4 outputs are available at http://www.acd.ucar.edu/wrf-chem/mozart.shtml. The program that generates CMAQ boundary conditions from MOZART-4 output is available at: http://www.camx.com/getmedia/a0c2d710-6187-47bb-af3f-bd9b4a4d19bb/mozart2camx-12jul17.tgz. The HERMESv2 outputs are available upon request. The modelled data in this study are available upon request from the corresponding author.

**Appendix A**

Table 1A shows the acronyms used in the text.





| BCON | Chemical boundary conditions to IP4, also named as imported $O_3$ contribution |
|------|-----------------------------------------------------------------------|
| BMA | Barcelona Metropolitan Area |
| CIP | Central Iberian Peninsula |
| CMAQ | Community Multiscale Air Quality model |
| EIP | Eastern Iberian Peninsula |
| EU12 | European domain at 12-km horizontal resolution |
| EV | Ebro Valley |
| GV | Guadalquivir Valley |
| IP | Iberian Peninsula |
| IP4 | Iberian Peninsula domain at 4-km horizontal resolution |
| ISAM | Integrated Source Apportionment Method |
| MDA8 | Maximum daily 8-hour average concentration |
| MMA | Madrid Metropolitan Area |
| MED | Mediterranean Sea |
| NEIP | Northeastern Iberian Peninsula |
| NIP | Northern Iberian Peninsula |
| NWIP | Northwestern Iberian Peninsula |
| SIP | Southern Iberian Peninsula |
| SNAP1 | Emission sector on combustion in energy |
| SNAP34 | Emission sector on combustion and processes in industry |
| SNAP7 | Emission sector on road transport, exhaust and non-exhaust |
| SNAP8 | Emission sector on non-road transport (international shipping) |
| WIP | Western Iberian Peninsula |
| WRF | Weather Research and Forecasting Model |

**Appendix B**

Definition of the discrete statistics used in the evaluation: correlation coefficient (Eq. B1), mean bias (Eq. B1) and root mean

5   squared error (Eq. B3). Where $C_m(x, t)$ and $C_o(x,t)$ are the modelled and observed concentrations at a location (x) and time

(t); N is the number of pairs of data. $\overline{C_m}$ and $\overline{C_o}$ are the modelled and observed mean concentrations over the whole period,

respectively.

$$r = \frac{\sum_{i=1}^{N}(C_m(x,t) - \bar{C}_m)(C_o(x,t) - \bar{C}_o)}{\sqrt{\sum_{i=1}^{N}(C_m(x,t) - \bar{C}_m)^2}\sqrt{\sum_{i=1}^{N}(C_o(x,t) - \bar{C}_o)^2}} \tag{B1}$$

$$MB = \frac{1}{N}\sum_{i=1}^{N}\left(C_m(x,t) - C_0(x,t)\right) \tag{B2}$$



$$RMSE = \sqrt{\frac{1}{N}\sum_{i=1}^{N}(C_m(x,t) - C_o(x,t))^2} \qquad\qquad (B3)$$

**Acknowledgments**

This study has been supported by the Spanish Ministry of Economy and Competitiveness and FEDER funds under the project PAISA (CGL2016-75725-R). This work was granted to access to the High Performance Computer resources of the "Red Española de Supercomputación" (AECT-2017-1-0008).  The views expressed in this article are those of the authors and do not necessarily represent the views or policies of the U.S. Environmental Protection Agency. Carlos Pérez García-Pando acknowledges long-term support from the AXA Research Fund, as well as the support received through the Ramón y Cajal programme (grant RYC-2015-18690) of the Spanish Ministry of Economy and Competitiveness.

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

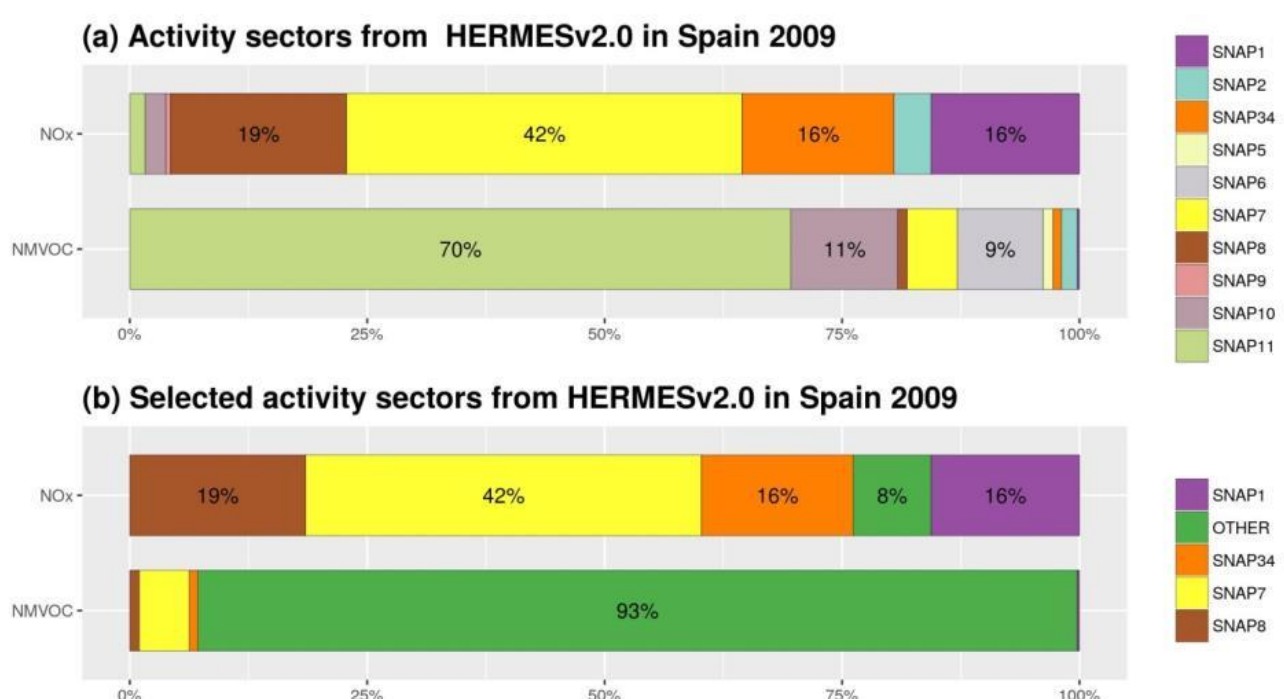

**Figure 1: Percentage of the contribution of emissions to total annual emissions by SNAP sector calculated by HERMES for Spain**
**2009 (a) and for the selected SNAP sector accounting for more than 90% of $NO_x$ total emission to be tracked with ISAM (b).**
**"OTHER" compiles the SNAP categories 2 (Residential combustion), 5 (Fugitive emissions from fuels), 6 (Solvent use), 9 (Waste**
**management), 10 (Agriculture) and 11 (Other sources).**





**Figure 2: (a) Temporal distribution of the MDA8 O₃ concentration during the extended summer (from April to September, AMJJAS) at the Spanish EIONET stations for the period 2000-2012 and the episode (from 21/07/2012 to 31/07/2012) by station type: IN (industrial), RB (rural background), TR (traffic) and UB (urban background). (b) Number of days exceeding the O₃ Target Value (120 μg/m³) by each day of the episode. (c) 90[th] percentile of the MDA8 O₃ concentration at the Spanish EIONET stations during the episode.**



**Figure 3: WRF-ARW meteorological fields at 6 UTC for July 25th, 28th and 31st in the EU12 domain: 10-m wind speed (W10, m/s), 2-m temperature (T2M, C), 6 h accumulated precipitation (Prec., mm), mean sea level pressure (MSLP, hPa), 500-hPa geopotential height in contours (Geo. Height, m), and 500-hPa temperature in shaded colors (T, ºC).**





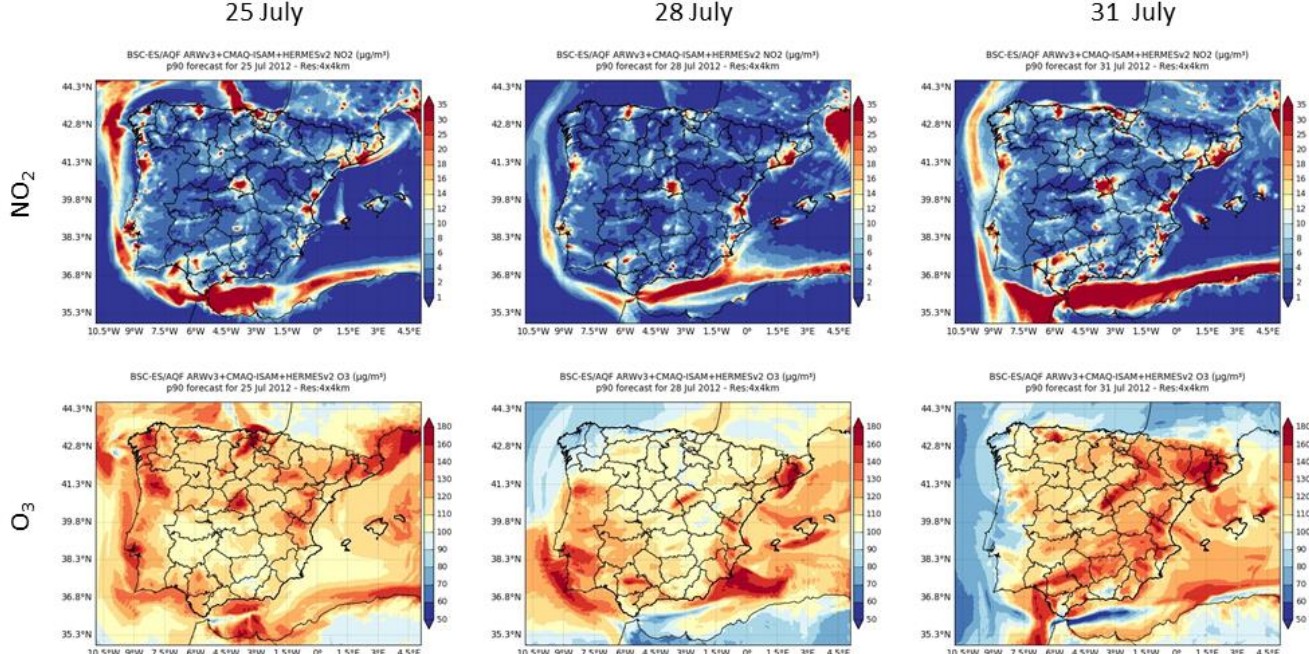

**Figure 4: Ground-based concentration maps (in µgm$^{-3}$) for NO$_2$ (first row) and O$_3$ (second row) corresponding to the 90$^{th}$ percentile of the hourly concentrations on 25$^{th}$ (first column), 28$^{th}$ (second column) and 31$^{st}$ (third columns) July 2012.**

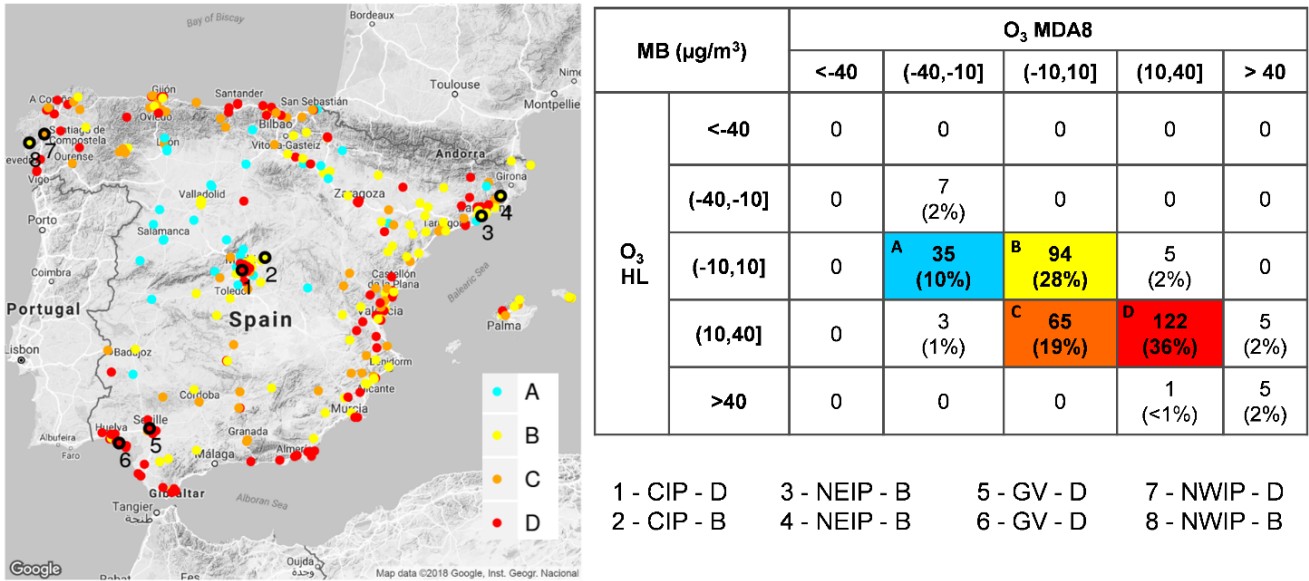

**Figure 5: Air quality stations classified by both mean bias (MB, in µg/m$^3$) for hourly and MDA8 O$_3$ at the Spanish EIONET stations and lumped by categories (A, B, C, and D). Numbered black circles indicate stations under study in Central IP (CIP; station 1 and 2), Northeastern IP (NEIP; station 3 and 4), Guadalquivir Valley (GV; station 5 6), and Northwestern IP (NWIP; station 7 and 8).**





Figure 6: Tagged O₃ concentrations (in µgm⁻³) corresponding to the 90th percentile (90p) of the hourly concentrations: SNAP1, SNAP34, SNAP7, SNAP8, OTHER, and BCON for July 25th (first column), 28th (second column) and 31st (third columns) in 2012.



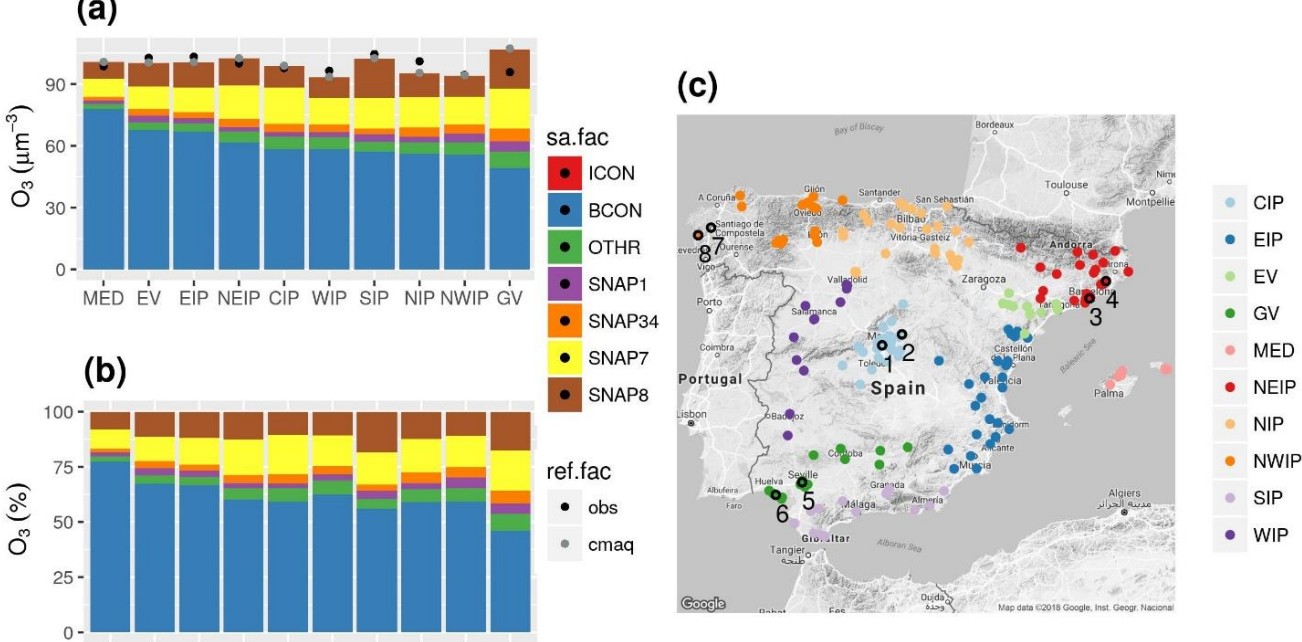

**Figure 7: Daily mean contribution in μgm⁻³ (a) and in percentage (b) of tagged sources to O₃ during exceedances of the observed 120 μgm⁻³ for MDA8 O₃ averaged by the identified receptor regions (c). Black and grey dots represent observed and modelled daily mean concentration during exceedances of 120 μgm⁻³ of the observed MDA8 O₃. Regions correspond to the Center of the IP (CIP), Eastern IP (EIP), Ebro Valley (EV), Guadalquivir Valley (GV), the Mediterranean Sea (MED), North-Eastern IP (NEIP), Northern IP (NIP), North-Western IP (NWIP), Southern IP (SIP) and Western IP (WIP). Numbered black circles indicate stations under study CI (1-2), NEIP (3-4), GV (5-6) and NWIP (7-8).**





**Figure 8: Source apportionment time series for O₃ and NO₂ concentrations (in µgm⁻³) in the episode at the selected stations in the Central IP (CIP) region (a,b), and in the North-Eastern IP (NEIP) )(c,d). Color bars (sa.fac) indicate the O₃ tags. Black and grey dots (ref.fac) indicate the observed and modelled concentrations, respectively. Black horizontal lines represent O₃ target value (120 µgm⁻³) and NO₂ limit value (40 µgm⁻³) as a reference. The location of the stations is shown in Fig. 7 by the corresponding numbers.**

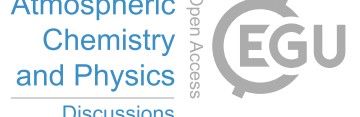

**Figure 9: Source apportionment time series for O₃ and NO₂ concentrations (in µgm⁻³) in the episode at the selected stations in the Guadalquivir Valley (GV) (a,b), and in the northwestern IP (NWIP) )(c,d). Color bars (sa.fac) indicate the O₃ tags. Black and grey dots (ref.fac) indicate the observed and modelled concentrations, respectively. Black horizontal lines represent O₃ target value (120 µgm⁻³) and NO₂ limit value (40 µgm⁻³) as a reference. The location of the stations is shown in Fig. 7 by the corresponding numbers.**





**Figure 9: Cross section of modelled mixing ratios (in ppb) for O₃ (a) and NO₂ (b) at a constant latitude (Latitude = 40.38°, Madrid city) of the daily Iberian Thermal Low circulation, equivalent to the conceptual scheme of Millán et al. (1996) for July 25th (first column), 28th (second column) and 30th (third column) at 06, 12 and 18 UTC. Dot lines indicate the boundaries of the planetary boundary layer height. Vertical arrows indicate the vertical wind. Up arrows depict positive winds. Note the different scale for the y-axis between O₃ and NO₂.**



**Table 1: Description of the O₃ tagged sources in the present study.**

| ISAM tag* | Emission by SNAP category | Description |
|---|---|---|
| SNAP1 | SNAP1 | SNAP1: Energy industry |
| SNAP34 | SNAP34 | SNAP34: Industry (combustion and processes ) |
| SNAP7 | SNAP7 | SNAP7: Road transport, exhaust and non-exhaust |
| SNAP8 | SNAP8 | SNAP8: Non-road transport (international shipping) |
| OTHR | SNAP2 + SNAP5 + | SNAP2: residential and commercial/institutional combustion |
| | SNAP6 + SNAP9 + | SNAP5: Fugitive emissions from fuels |
| | SNAP10 + | SNAP6: Product use including solvents |
| | SNAP11 | SNAP9: waste management |
| | | SNAP10: Agriculture |
| | | SNAP11: Other sinks |
| BCON | - | Chemical boundary conditions to IP4 domain from the EU12 simulation which includes the contribution from Europe and international contribution from MOZART-4. O₃ external contribution |
| ICON | - | Initial chemical condition of the domain IP4 |

*Each ISAM tag is applied to O₃ and its precursor species in the CB05 (NO$_x$ and VOCs). NO$_x$ species contributing to O₃ formation involve (9 species): NO, NO$_2$, nitrogen trioxide (NO$_3$), dinitrogen pentoxide (N$_2$O$_5$), nitrous acid (HONO), peroxyacyl nitrates (PAN), higher peroxyacyl nitrates (PANX), peroxynitric acid (PNA), and organic nitrates (NTR). VOC species contributing to O₃ formation include (14 species): acetaldehyde (ALD2), higher aldehydes (ALDX), ethene (ETH), ethane (ETHA), ethanol (ETOH), formaldehyde (FORM), internal olefin (IOLE), isoprene (ISOP), methanol (MEOH), olefin (OLE), paraffin (PAR), monoterpene (TERP), toluene (TOL), and xylene (XYL).



**Table 2: Statistics for hourly O₃, MDA8 O₃, and hourly NO₂ concentrations in the episode as a function of the station type. Exceedances indicate the number of exceedances of the European Air Quality Directive Standards for hourly O₃ (180 µgm⁻³), MDA8 O₃ (120 µgm⁻³) and hourly NO₂ (200 µgm⁻³). N indicates the number of monitoring stations used in the statistics calculation. MO and MM depict the measured and modelled mean concentrations, respectively. Statistics are calculated by considering more than 75 % of the hours in a day, as established by Directive 2008/50/EC. The statistics correspond to following quantiles 50ᵗʰ (25ᵗʰ, 75ᵗʰ) by station. Type indicates the station categories in the calculation of statistics: all the stations (ALL), industrial (IN), traffic (TR), urban background (UB) and rural background (RB) stations. Note that there are stations that do not fit in any of these four classifications.**

| Pollutant | Type | Exceedances | N | MO ($\mu gm^{-3}$) | MM ($\mu gm^{-3}$) | MB ($\mu gm^{-3}$) | RMSE ($\mu gm^{-3}$) | r |
|---|---|---|---|---|---|---|---|---|
| Hourly O₃ | ALL | 26 | 347 | 77.3 (66.7,86.5) | 89.9 (83.5,95.6) | 12.6 (4.4,20.2) | 26.7 (21.5,32.1) | 0.6 (0.6,0.7) |
| | IN | 5 | 106 | 74.1 (62.2,83.2) | 82.2 (83.4,94.5) | 14.1 (4.5,21.5) | 26.8 (21.1,32.8) | 0.7 (0.6,0.7) |
| | TR | 0 | 70 | 68.4 (57.1,76.7) | 83.8 (74.4,89.2) | 15.9 (8,21.8) | 28.8 (24.5,33.9) | 0.6 (0.5,0.7) |
| | UB | 0 | 56 | 74.8 (64.1,80.0) | 89.4 (81.4,94.5) | 15.6 (7.8,21.3) | 28.2 (24.5,33.5) | 0.7 (0.6,0.7) |
| | RB | 17 | 66 | 91.2 (80.9,97.1) | 96.2 (92.6,99.6) | 4.5 (-3.3,14.7) | 21.2 (18.3,28.0) | 0.7 (0.6,0.7) |
| MDA8 O₃ | ALL | 751 | 347 | 101 (91.0,113.4) | 106.2 (101.8,113.2) | 5.7 (-3.7,16.9) | 17.9 (13.5,25.4) | 0.6 (0.4,0.8) |
| | IN | 204 | 106 | 96.7 (88.9,109.5) | 104.8 (99.4,109.2) | 5.7 (-2.9,16.5) | 17.0 (13.0,24.5) | 0.7 (0.5,0.8) |
| | TR | 62 | 70 | 92.7 (83.6,99.9) | 104.1 (98.3,110.2) | 13.3 (4.6,23.8) | 21.2 (15.2,31.2) | 0.5 (0.3,0.8) |
| | UB | 86 | 56 | 100.0 (89.3,105.5) | 107.6 (102.2,122.8) | 14.1 (2.1,21.7) | 22.6 (15.6,28) | 0.6 (0.3,0.8) |
| | RB | 260 | 66 | 113.1 (104.9,119.9) | 108.6 (104.5,113.2) | -3.7 (-11.1,5.5) | 15 (11.8,20.2) | 0.7 (0.6,0.8) |
| Hourly NO₂ | ALL | 3 | 357 | 14 (8.4,21.2) | 9.9 (4.4,16.2) | -4.1 (-8.4,-0.5) | 12.8 (8.2,17.6) | 0.4 (0.3,0.6) |
| | IN | 0 | 120 | 11.5 (7.9,16.3) | 8.8 (3.8,14.7) | -3.0 (-6.6,0.2) | 11.0 (7.9,15.6) | 0.4 (0.3,0.5) |
| | TR | 3 | 95 | 22.9 (17.7,28.7) | 14.1 (8.7,21.6) | -7.4 (-13.9,-2.6) | 18.4 (14.1,23.1) | 0.4 (0.3,0.6) |
| | UB | 0 | 63 | 17.0 (13,21.2) | 13.2 (7.8,18.4) | -4.0 (-8,-0.3) | 14.1 (10.5,17.3) | 0.5 (0.4,0.6) |
| | RB | 0 | 41 | 4.3 (3.5,9.3) | 2.8 (1.6,4.0) | -2.0 (-5.3,-0.3) | 4.4 (2.7,7.1) | 0.3 (0.2,0.4) |