# Peer review of "Ozone source apportionment during peak summer events over southwestern Europe"

_Atmospheric Chemistry and Physics, 2018_

## Referee Comment (RC1) · Anonymous Referee #2 · 16 Oct 2018

General comments: The manuscript "Ozone source apportionment during peak summer events over southwestern Europe" presents a source apportionment study based on the application of an Eulerian chemical-transport model for the Iberian Peninsula for a specific summer episode in 2012. The research is interesting and it may constitute a valuable contribution towards a better understanding of O3 dynamics and potential options to improve ground-level concentration values in that region. Nonetheless, the outcomes of their research may potentially be of relevant for other areas and thus, of interest for a wider community. The paper is clearly structured and well written but before it can be considered for publication in Atmospheric Chemistry and Physics there are some obscure points and potential flaws that should be carefully addressed to guarantee that the results and their interpretation are correct. Some specific suggestions to

do so are provided below.

Specific comments Introduction In the introduction the authors provide a good background of the ozone related issues in the Iberian Peninsula including an overview of meteorological conditions typically associated to high ground-level O3 concentrations along with a number of relevant references. They also briefly discuss the trends and justify the need for their research. The topic is timely and interesting not only from the scientific point of view but also considering the legal implications and the need to identify potential interventions that may help alleviate O3 pollution in the Mediterranean Basin. Given the limitations of brute force methods for source apportionment studies of secondary pollutants, the authors apply the Integrated Source Apportionment Method (ISAM) implemented within the Community Multiscale Air Quality (CMAQ) with 4x4 km2 resolution for the whole Iberian Peninsula during a 10-day specific episode in summer 2012. The rationale and approach is clear but it would be interesting to explicitly state the main purpose of the study since this is relevant to understand whether the experiment design is appropriate and what are the limitations that can be expected from potential conclusions.

P3.Line 18: when discussing the possible reasons for the observed increase of O3 in some urban areas in the Iberian Peninsula, the authors assume a VOC-limited situation. Reductions on NOX emissions have necessarily reduced NO titration but I suggest them to remove that assumption regarding the VOC-NOX regime because it may be an oversimplification and not necessarily true in all cases/seasons.

P4.Line 29: the authors claim that the period analysed (between July 21st and 31st) is representative of typical summer synoptic conditions in that particular region. That is quite a strong statement and it would require substantial discussion and evidence to demonstrate to what extent that is true. Regardless of that, although the period may be characteristic in terms of synoptic conditions, O3 dynamics as previously stated, is strongly conditioned by both long-range transport and local conditions, including emissions of O3 precursors, initial chemical conditions, etc. Although the paper constitutes

a valuable contribution to the understanding of O3 pollution in the Iberian Peninsula, I think that it cannot be assumed that the outcomes of the study may provide a source apportionment comprehensive description. Consequently, the insight to support the design abatement policies is limited and caution should be used to avoid extracting incorrect conclusions.

Methodology: An illustrative description of the CALIOPE and HERMES systems is provided in section 2.1. along with a number of relevant references. The authors state, however that emissions are based on 2009 since that is the most recent year with updates information on local emission activities. That statement is hard to understand and it may deserve further clarification. In addition, this section would benefit from a more consistent discussion on how this methodological choice may impact the results and to what extent potential inconsistencies with meteorology and boundary conditions for that specific modelling period are compatible with the research specific aim (which is not clearly identified). This issue may be acceptable to gain a general understanding of O3 contribution over a long period of time but for a short, high-O3 episode this may be a potential flaw that should be carefully addressed. VOC emissions are particularly relevant input for this analysis. However, our current understanding of VOC emission is limited, especially in urban areas (Lewis, 2018) which makes it difficult to accurately apportion contributions to tropospheric O3. It is also widely accepted that biogenic VOCs play a major role on atmospheric photochemistry. For this study, the authors rely on the Model of Emissions of Gases and Aerosols from Nature (MEGAN) version 2.04 (P5.Line 30). This version was revised and extended through version 2.1 (Guenther et al., 2012) that also includes some code fixes. I'd strongly suggest the authors to perform a sensitivity run to understand whether using an outdated version of MEGAN may introduce relevant biases into their simulation. Species tagging and emission categories selection described in section 2.3 seem sensible although VOC emission shares should be reviewed taking into account the previous comment. I'm also concerned about a potential double counting and/or erroneous spatio-tempral allocation of VOCs emissions from agriculture since plant functional types considered in

MEGAN include crops. From previous literature, a share of 70% of total VOCs from SNAP 11 (nature) seems too high and may support that shortcoming. Please, double check this potential issue since it may bring about a considerable bias the outcomes of the study. The study makes advantage of a remarkably dense network of monitoring sites in the area of study to assess model performance through the computation of a series of common statistics (appendix B). The results however are difficult to interpret in their current form. Please, see corresponding suggestion in the results section. Refs: -Lewis, A. C. The changing face of urban air pollution. Science (New York, N.Y.), 16 February 2018, Vol.359(6377), pp.744-745. DOI: 10.1126/science.aar4925 -Guenther, A. B., Jiang, X., Heald, C. L., Sakulyanontvittaya, T., Duhl, T., Emmons, L. K., and Wang, X.: The Model of Emissions of Gases and Aerosols from Nature version 2.1 (MEGAN2.1): an extended and updated framework for modeling biogenic emissions, Geosci. Model Dev., 5, 1471-1492. DOI:10.5194/gmd-5-1471-2012, 2012

Results: The authors discuss that the episode at hand concentrated an important percentage of exceedances in Spain (if I understood correctly; the first paragraph may be reviewed for the sake of clarity). However, the inspection of panel a) in Fig 2. does not seem to indicate that this episode was particularly severe since the distribution of MDA8 is not dissimilar to those of previous years, even when they reflect the concentrations over a 6 month period. In addition, the outliers apparently have moderated values. It would be interesting to make the point for such comparison. I'm not completely sure that it is sound to identify a high pollution episode at national level since O3 largely depends on regional features. If O3 levels were actually high all over the modelling domain, the influence of exported ozone (influenced by synoptic conditions) may be too high and thus, the representativeness of the results and the potential implications policy-wise, rather limited. Please, reflect on that. They also claim that the episode affected the central and north of the IP, but Fig. 2c shows high concentration spots all over the domain (or maybe the colour scale is not clear enough). It is also hard to see what the influence of Madrid and Barcelona plumes is (something consistent, to my understanding, with dominant stagnant conditions). Please, try to clarify. Fig 3 illustrate the meteorological conditions through some WRF -ARW outputs. I understand that a thorough model evaluation is not the main purpose of the paper but a minimal check (also through a statistic evaluation) of the credibility of the meteorological simulation (mainly wind fields) may substantially help the authors to gain a better insight of their results. For instance, that would help them to contrast hypotheses such us the one made in P10.Line 25. and following, where they attribute O3 nighttime overestimation to the underestimation of vertical mixing during nighttime stable conditions. In that case, a comparison of observed (where available) and modelled PBLH may be useful. As for the discussion of the general meteorological conditions, I'm not sure that the attempt to discriminate between ITL and NWadv situations is nor illustrative or needed. The discussion is hard to follow and the application of deterministic CTM may make that effort redundant.

The results of the statistical evaluation for CMAQ outputs are summarized in Table 2. Some suggestions for this table: - Please include in the caption whether the exceedance column refers to observed or modelled values (the missing one may be also included either way) - Two or three decimal points for r may be used - MNB (%) may be more illustrative than MB (that can be derived directly from MM – MO) - It may be misleading to pool together the statistics for different types of monitoring stations. As the authors discuss, it is arguable that outputs from a 4x4 km2 model exercise should be compared against observations at traffic locations - It is unclear why some monitoring sites wouldn't fit into any of the categories considered. Please elaborate and state the rationale to include them in the analysis It may be interesting to put these results into perspective by comparing them with those from other modelling exercises based on similar model suites in the IP or elsewhere (besides referring to previous applications of CALIOPE itself).

P11.L10-14. The description of the performance-based categories is hard to follow and it is already condensed in Fig. 5. Please, simplify or simply remove that passage. I'd suggest the authors to re-compute statistics and assessment with the alternative

BVOC emissions model run mentioned earlier. If mention to specific cities or areas is made (e.g. Ebro Valley, Lleida Plain), please identify them in any of the maps in the manuscript.

I'd generally advice the authors to condense the discussion and try to highlight their main findings not to unnecessarily extend this section. The information summarized in Fig. 7 is interesting although the concept of "receptor regions" is unclear. It seems reasonable in terms of geographical location but differences regarding contributions from different sectors are not evident, especially if the results are put into the perspective of the typical uncertainties of modelling exercises that can be inferred from the model evaluation previously presented. It is interesting noting a relatively large contribution from the SNAP 8 sector. The share of mobile sources is particularly important in the SIP area, which would be consistent with the discussion regarding the influence of shipping. However, other areas such as GV or even CIP present a non-negligible contribution. Could the authors elaborate on that? Maybe the discussion in section 3.4 is too profuse and should be substantially shortened. I encourage the authors to summary here their findings and provide the region-by-region discussion as supplementary material, including Fig. 8 and Fig. 9. Oppositely, the rationale for the station sub-set selection may deserve further explanation. In general, the section is abundant in hypotheses and subjective interpretation that are not clearly supported by evidence. Personally, I don't think this contribution really benefits from such approach. The paper may be restricted to a more solid and consistent analysis and discussion of the findings from the application of CMAQ-ISAM. That is novel and interesting enough and further attempts to relate the results with detailed regional dynamics and atmospheric patterns may very well be addressed in future specific studies (using more specific methods and data, e.g. better resolved emission inventories).

Discussion and conclusions: The authors claim that the modelling exercise presented allowed an in-depth evaluation of the modelling system applied. This relates to my last comment regarding the results section. The paper may lack a well-defined objective

and presents a huge amount of information without a clear purpose. For example, if the main interest was to assess the modelling system capabilities and identify options for improvement, the results and analysis should gravitate towards a more detailed statistical analysis within a better defined methodological framework. I acknowledge a valuable study but I think the authors should revise their manuscript under a clearly defined scientific question avoiding an excessive spread in their discussion that may lead to inconsequential or cursory analyses and reflect that also in this section. As for the discussion on model uncertainty I find particularly important to take into account the observations regarding biogenic emissions, although the mismatch between emissions and meteorological conditions may also hinder the discriminating power of the results. In any case, caution should apply since the timespan of the period analysed makes it difficult to extract general conclusions. This is particularly important for the regulatory implications that may be derived from this study. As the authors conclude, I find reasonable to base recommendations for abatement strategies in more specific, regional scale, detailed analyses. Consequently I'd keep such conclusions to a minimum in this contribution.

Technical issues and typos: Please revise equations 1 to 4 for a better readability P7.Line 4: SNAP3 and 4 P8.Lines 12-13: the brackets are not needed P11.Line 11: "for 93% of the stations" instead of "for the 93% of stations" P12.Line 2: "extremely low winds"? P20.Line 8: "O3at" is missing a space P34.Line 5 (Fig. 2 caption): I guess the authors mean "Number of stations" instead of "Number of days"

---

## Referee Comment (RC2) · Anonymous Referee #3 · 29 Dec 2018

The paper gives important contribution to address the source apportionment study regarding ozone episode occurred in Spain. The paper is well structured and presents a complete analysis of the modelling results. However, there are some major points that should be addressed before recommended for publication. Besides that, English should be revised along the manuscript, there is some inconsistencies and grammatical errors. See below major and minor comments.

Major changes - Abstract; Line 15 (Page 4/Line 2): there is recently studies that showed that source-apportionment methods are not adequate to investigate plans and mitigation measures, in particular for non-linear pollutants like ozone, and that for that purpose "scenario analysis" based on "brute-force" are recommended. Authors should revise the text along the manuscript where it is mentioned the purpose of "designing

[Figure]

plans", which should not be the final objective of this source-apportionment study. (see Clappier, A., Belis, C., Pernigotti, D., Thunis, P. Source apportionment and sensitivity analysis: two methodologies with two different purposes. Geosci. Model Dev. Discuss. 10, 4245-4256 (2017)). - Page 4/Line 7-9: please review this sentence according to what has been commented before - Page 5/Line 26: the authors should comment about the representativeness of the 2009 emissions to the 2012 SA study presented. From 2009 to 2012 several changes happened in society and economy which was reflected in emission data. - Page 7, Line 9: SNAP2 activity can be a particular important source for ozone precursors. Authors should comment about that when they mentioned that SNAP2 is aggregated with other activity sectors. - Page 7, Lines 25-30: in the scope of FAIRMODE – Forum for Air Quality Modelling in Europe – tools were developed, namely the DELTA-Tool, to evaluate air quality models and conclude about their suitability to be used for legislation purposes. The authors should consider the application of this tool to evaluate model performance instead of calculating the traditional statistical indicators. In any case, authors should justify why they decided not using this tool. - Page 13, Lines 16-17: this information (model performance in terms of o3 peaks) should be presented and discussed in the model validation section - Page 13, Line 22: this sentence should be completed with information about the area where this impact (up to 8%) is verified. - Page 15, Lines 4-5: The authors should quantify how "model reproduces reasonably well" - Please clarify the sentence "The NO2 overestimation correlates with the highest road transport contribution" - Please explain why: "The results point towards a poor representation of the vertical mixing during the stagnant conditions" - Page 16, Lines 4-10: this should be placed in the model validation section - Page 16, Lines 13-14: how can the authors conclude that the model is able to reproduce all these different processes? Can the authors support better this statement? - Page 18, Line 4: since only one rural station is analysed, the authors should not generalize as "In rural background areas. . ." - Page 18, Lines 30-33 to Page 19: the authors analyse the vertical profile in a single point, but this will be not representativeness of the all study domain. Authors should change the text according to this limitation and

comment it, or increase the number of points analysed. - Figure 2: please review the figure caption "Number of days exceeding the O3 target value (120 ug.m-3) by each day of the episode"

Minor changes: - Page 1/Line 20: write 4x4 km2 instead of 4x4 km (please correct this along the manuscript) - Authors should refer the modelling system (CALIOPE) in the abstract - Page 3, Line 4: The following reference should be added, since it is the biggest ozone episode occurred in IP region: "Monteiro A., Gama C., Candido M., Ribeiro I., Lopes M. (2016) Investigating ozone high levels and the role of sea breeze on its transport. Atmospheric Pollution Research 7, 339-347. - Page 7, Line 23: please indicate how many stations measure both O3 and NO2 pollutants - Page 10, Lines 17, 24, 29: please add "average" when mentioning "hourly O3" (the values presented are an average of different locations and not an "hourly O3 data" - Page 12, Lines 2-7: the following reference should be added to support this part: Borrego C., Monteiro A., Martins H., Ferreira J., Fernandes A.P., Rafael S., Miranda A.I., Guevara M., Baldasano J.M. (2016). Air quality plan for ozone: a case-study for North Portugal. Air Quality, Atmosphere & Health 9 (5), 447–460. - Page 14, Line 30: please review the English - Page 17, Lines 19-24: authors should consult and use the following reference that compares the different shipping emission inventories mentioned: Russo M.A., Leitão J., Gama C., Ferreira J., Borrego C., Monteiro A. (2018) Shipping emissions over Europe: a state-of-the-art and comparative analysis. Atmospheric Environment 177, 187–194. - Page 18, Line 22: please replace "These O3. . ." by "The results presented before. . ." - Figure 3: Please review the units used along the manuscript, like "m/s"

---

## Author Comment (AC1) · 25 Feb 2019

We thank the Referee #2 for his/her thorough and constructive comments and suggestions, which have contributed to improve the quality of our paper. All his/her comments have been implemented and commented accordingly in the revised version of the manuscript.

Please, find below the item-by-item response. For more details on the review process, we have uploaded the manuscript with track-changes.

1. Reviewer #2: In the introduction, the authors provide a good background of the ozone related issues in the Iberian Peninsula including an overview of meteorological conditions typically associated to high ground-level O3 concentrations along with a

number of relevant references. They also briefly discuss the trends and justify the need for their research. The topic is timely and interesting not only from the scientific point of view but also considering the legal implications and the need to identify potential interventions that may help alleviate O3 pollution in the Mediterranean Basin. Given the limitations of brute force methods for source apportionment studies of secondary pollutants, the authors apply the Integrated Source Apportionment Method (ISAM) implemented within the Community Multiscale Air Quality (CMAQ) with 4x4 km2 resolution for the whole Iberian Peninsula during a 10-day specific episode in summer 2012. The rationale and approach is clear but it would be interesting to explicitly state the main purpose of the study since this is relevant to understand whether the experiment design is appropriate and what are the limitations that can be expected from potential conclusions.

Authors: We thank the Reviewer for their assessment of the scope and methodology of the manuscript. We have rewritten some parts of the abstract and the introduction to clarify the main propose of the study as follows:

In the abstract (Page 1 – Line 19-21): "The main goal of this study is to provide a first quantitative estimation of the contribution of the main anthropogenic activity sectors to peak O3 events in Spain relative to the contribution of imported (regional and hemispheric) O3. We also assess the potential of our source apportionment method to improve O3 modelling"

In the introduction (Page 4 – Line 25-34): "The integrated source apportionment tools combined with high-resolution emission and meteorological models can help unravelling the sources responsible for peak summer events of O3 in the Western Mediterranean Basin. Quantifying the contribution of emission sources during acute O3 episodes is a prerequisite for the design of future mitigation strategies in the region. In this framework, the main goal of this study is to provide a quantitative estimation of the contribution of the main anthropogenic activity sectors compared to the imported concentration (regional and hemispheric) to peak O3 events in Spain. We also assess the potential of our source apportionment method to improve O3 modelling. Our study applies for the first time a countrywide O3 source apportionment at high resolution over the Iberian Peninsula during the period between July 21st and 31st, 2012. We use the CMAQ-ISAM within the CALIOPE air quality forecast system for Spain (www.bsc.es/caliope), which runs at a horizontal resolution of 4x4 km2 over the IP. The system is fed by the HERMESv2.0 emission model, which provides disaggregated emissions based on local information and state-of-the-art bottom-up approaches for the most polluting sectors."

2. Reviewer #2: P3.Line 18: when discussing the possible reasons for the observed increase of O3 in some urban areas in the Iberian Peninsula, the authors assume a VOC-limited situation. Reductions on NOx emissions have necessarily reduced NO titration but I suggest them to remove that assumption regarding the VOC-NOx regime because it may be an oversimplification and not necessarily true in all cases/seasons.

Authors: We agree with the reviewer. We have removed the assumption regarding the VOC-NOx regime as follows:

Page 3–Line 17-19: "The reasons behind the urban O3 upward trend are not clear yet due to the complex VOC-NOx regime; part of the O3 increase may have resulted from the reduction of NO emissions relative to NO2 and therefore to a lower NO titration effect in VOC-limited situations."

3. Reviewer #2: P4.Line 29: the authors claim that the period analysed (between July 21st and 31st) is representative of typical summer synoptic conditions in that particular region. That is quite a strong statement and it would require substantial discussion and evidence to demonstrate to what extent that is true. Regardless of that, although the period may be characteristic in terms of synoptic conditions, O3 dynamics as previously stated, is strongly conditioned by both long-range transport and local conditions, including emissions of O3 precursors, initial chemical conditions, etc. Although the paper constitutes a valuable contribution to the understanding of O3 pollution in the

Iberian Peninsula, I think that it cannot beassumed that the outcomes of the study may provide a source apportionment comprehensive description. Consequently, the insight to support the design abatement policies is limited and caution should be used to avoid extracting incorrect conclusions.

Authors: Our characterization of the study period (July 21st and 31st, 2012) is based on the circulation type classification performed in Valverde et al. (2014, Circulation-type classification derived on a climatic basis to study air quality dynamics over the Iberian Peninsula. Int. J. Climatol. 35 (8)). Specifically, the classification in Valverde et al. (2014) is designed to study air quality dynamics over the Iberian Peninsula using an objective synoptic classification method over the present climate (1983–2012). According to the classification in Valverde et al. (2014), our study episode starts with an Iberian Thermal Low (ITL) (21-25th July, 2012), followed by a northwestern advection from the Atlantic (NWad) (26-29th July, 2012) and finishing with another ITL (30-31st July, 2012). ITL and NWad are circulation types that typically affect the Iberian Peninsula, which represent the 44% of the days in the IP both taking place in summer and alternate each other (Valverde et al., 2014).

We have provided a summary of these evidences to support the representativeness of the episode in the revised version of the manuscript as follows:

Page 9–Line 25: "Our characterization of the study period is based on the circulation type classification performed in Valverde et al. (2014), who developed an objective synoptic classification method over the period 1983–2012, specifically designed to study air quality dynamics over the IP. [...]. According to the circulation type classification in Valverde et al. (2014), the selected episode started with the development of the ITL (July 21st-25th), followed by a NWad-venting period (July 26th-29th) and ended with the development of another ITL (July 30th-31st)."

On the other hand, we want to remark that our study is a first quantitative O3 source apportionment study, and the representativeness of our results is limited because they

are focused in just one episode. We have clarified this limitation in the manuscript in the Section 4 as follows: Page 20–Line 1: "Our study has provided a first estimation of the main sources responsible for high O3 concentration in the Western Mediterranean Basin during the period July 21st -31st, 2012."

Page 22–Line 22-23: "[. . .] future studies should preferentially cover multiple summer periods in order to improve representativeness."

Furthermore, we want to highlight that the main goal of this study is not the design abatement policies, but it is to provide a first quantitative estimation of the contribution of the main anthropogenic activity sectors to peak O3 events in Spain relative to the contribution of imported O3. Actually, source apportionment techniques alone cannot be used to the design abatement policies. Subsequent source sensitivity analyses tailoring the identified main contribution sources could predict how O3 will respond to reductions in precursor emissions. Both, source apportionment and source sensitivity are complementary and essential studies to define the most efficient O3 abatement strategies in the Western Mediterranean Basin. Therefore, this study has provide a perspective about the potential use of source apportionment methods for regulatory studies in non-attainment regions as a prerequisite for the design of future mitigation strategies. We have added some remarks about this point as follows:

Page 4 – Line 28-30 (Section 1. Introduction): "In this framework, the main goal of this study is to provide a first quantitative estimation of the contribution of the main anthropogenic activity sectors compared to the imported concentration (regional and hemispheric) to peak O3 events in Spain."

Page 22–Line 23-27 (Section 4. Discussion and conclusions): "We note that our results cannot predict whether emission abatement will have either a positive or negative effect in O3 changes due to the non-linearity of the O3 generation process. Subsequent source sensitivity analyses tailoring the identified main contribution sources could predict how O3 will respond to reductions in precursor emissions, which are essential to

define the most efficient O3 abatement strategies in the Western Mediterranean Basin."

Page 23-Line 4-7 (Section 4. Discussion and conclusions): "This work has quantified the local and imported contributions to O3 during an episode in a particular area in sourthwestern Europe. In addition, we have provided a perspective about the potential use of source apportionment method for regulatory studies in non-attainment regions. Further O3 source apportionment studies targeting other nonattainment regions in Europe are necessary prior to design local mitigation measures that complement national and European-wide abatement efforts."

4. Reviewer #2: An illustrative description of the CALIOPE and HERMES systems is provided in section 2.1. along with a number of relevant references. The authors state, however that emissions are based on 2009 since that is the most recent year with updates information on local emission activities. That statement is hard to understand and it may deserve further clarification. In addition, this section would benefit from a more consistent discussion on how this methodological choice may impact the results and to what extent potential inconsistencies with meteorology and boundary conditions for that specific modelling period are compatible with the research specific aim (which is not clearly identified). This issue may be acceptable to gain a general understanding of O3 contribution over a long period of time but for a short, high-O3 episode this may be a potential flaw that should be carefully addressed.

Authors: Our methodological choice has been to use a detailed bottom-up emission inventory instead of a typical top-down regional emission inventory. Bottom‐up emissions, estimated using source‐specific emission factors and activity statistics, accurately characterise pollutant sources and allow obtaining more realistic results than the ones reported by top-down or regional emission inventories. However, they require very large efforts to be compiled and consequently the updating processes cannot be implemented year-to-year.

In HERMESv2 emissions are based on 2009 data, which was the closest year with updated information on local emission activities in HERMES at the time this work started.

To understand the impact of the use of 2009 data to study year 2012 we revised the EMEP Centre on Emission Inventories and Projections (EMEP-CEIP), which collects and reviews the national emission inventories from Parties to the Convention on Long-range Transboundary Air Pollution (CLRTAP). Between 2009 and 2012 total NOx and NMVOC emissions in Spain decreased by -10.6% and -10.7%, respectively (EMEP CEIP, 2019). For NOx, around 80% of this reduction is linked to a reduction of road transport emissions, whereas in the case of NMVOC ~50% of the reduction is due to a decrease of industrial emissions. NOx emissions from shipping in Europe have also decreased in the period 2009-2012 by 15%.

For our modelling study, we consider these differences as small and acceptable, and not creating any major inconsistency. The difference of 10-15 % in emissions for certain precursors between 2009 and 2012 is within the typically larger ranges of uncertainty in emission inventories. We also note that all our results are thoroughly evaluated and critically assessed using observations.

In any case, we have followed the reviewer's suggestion, and we have discussed in the manuscript the potential impact of these differences when the contribution of each emission sector is analysed:

Page 17–Line 32-33: "[. . .] This factor, added to the 15% decrease of NOx shipping emissions observed in Europe between 2009 (HERMESv2.0 base year) and 2012 (EMEP CEIP, 2019) could explain the discrepancies observed."

Page 18–Line 12-14: "[. . .] it has been estimated that between 2009 and 2012 energy production in coal-fired power plants increased from 13.1% to 19.4% (UNESA, 2012), which implied an increase of NOx emissions from the power industry sector of around 19.5% (EMEP CEIP, 2019)."

The Section 4 of the revised version of the manuscript includes now a comment on

the methodological implication of using 2009 emissions for O3 source apportionment studies in an episode in 2012 as follows:

Page 5-Line 31-32: "HERMESv2.0 is currently based on 2009 data, which is the closest year with updated information on local emission activities in HERMES at the time this work started."

Page 21-Line 31: "Our methodological choice has been to use a detailed bottom-up emission inventory instead of a typical top-down regional emission inventory. Bottom‐up emissions, estimated using source‐specific emission factors and activity statistics, accurately characterise pollutant sources and allow obtaining more realistic results than the ones reported by top-down or regional emission inventories. To understand the impact of the use of 2009 data to study year 2012, we revised the EMEP Centre on Emission Inventories and Projections (EMEP-CEIP), which collects and reviews the national emission inventories from Parties to the Convention on Long-range Transboundary Air Pollution. Between 2009 and 2012, total NOx and NMVOC emissions in Spain decreased by -10.6% and -10.7%, respectively (EMEP CEIP, 2019). For NOx, around 80% of this reduction is linked to a reduction of road transport emissions, whereas in the case of NMVOC ∼50% of the reduction is due to a decrease of industrial emissions. For our modelling study, we consider these differences as small and acceptable, and not creating any major inconsistency. The difference of 10-15 % in emissions for certain precursors between 2009 and 2012 is within the typically larger ranges of uncertainty in emission inventories."

Reference:

EMEP CEIP, 2019. Officially reported emission data. Available at: http://www.ceip.at/ms/ceip_home1/ceip_home/data_viewers/official_tableau/ (last access February 2019)

5. Reviewer #2: VOC emissions are particularly relevant input for this analysis. However, our current understanding of VOC emission is limited, especially in urban areas

(Lewis, 2018) which makes it difficult to accurately apportion contributions to tropospheric O3.

Authors: For the estimation of the main VOC contributors in Spanish urban areas, namely the road transport (SNAP07) and the use of solvents (SNAP06), the HERMESv2.0 emission model uses a combination of bottom-up approaches and downscaling methodologies. The sectors road transport and solvents together account for more than 80% of total VOC emissions in Barcelona and Madrid cities (Soret et al., 2014).

For traffic, HERMESv2.0 estimates VOC emissions according to the Tier 3 method described in the EMEP/EEA guidelines, which is fully incorporated in COPERT 4. Speciation factors to map total VOC to the CB05 chemical mechanism species are also obtained from the EMEP/EEA guidelines.

For the solvent sector, emissions are estimated performing a downscaling methodology of the original Spanish National Emission Inventory due to the lack of specific information on activity data and emission factors. The Spanish inventory, developed by the Spanish Ministry of the Agriculture, Food and Environment (MAPAMA, personal communication), represents the official Spanish contribution to the EMEP emission inventory. It reports total annual emissions of primary pollutants by NUTS 2 level and SNAP elemental activity and it is based on the EMEP/EEA guidelines combined with local activity data (see the corresponding Inventory Informative Report, IIR, for more information: http://www.ceip.at/ms/ceip_home1/ceip_home/status_reporting/2018_submissions/).

HERMESv2 assigns a specific spatial proxy, temporal and speciation profile to each pollutant activity after defining it as point, lineal or area source. The speciation treatment of the original emissions is done using the profiles reported by the SPECIATE database (https://www.epa.gov/air-emissions-modeling/speciate-version-45-through-40). More details on the methods used to estimate VOC emissions from

these two sectors can be found in Guevara et al. (2013).

Despite the efforts in providing detailed emission input data, it is true that there are still a lot of uncertainties and room for improvement in the estimation of urban VOC emission inventories, as stated recently by several works (Liu et al., 2017; McDonald et al., 2018; Lewis, 2018). Most of these uncertainties are due to the lack of in-situ observational data, which is key for the development of local speciation profiles and for the evaluation of modelled concentrations of VOC. In order to overcome this problem, continuous monitoring of urban VOC should be performed in Spanish cities, following the example of other regions in which O3 is also a major problem such as Mexico City (Jaime-Palomera et al., 2016). On the other hand, the use of satellite observations of formaldehyde (HCHO) columns to constrain urban VOC emissions could be also pointed out as a future task to improve the representativeness of urban emission inventories (Zhu et al., 2014).

Urban VOC emissions are particularly relevant emissions for this analysis, and uncertainties in the estimation of urban VOC emission inventories makes also uncertain their contribution to tropospheric O3 source apportionment studies.

In the reviewed version of the manuscript, we have added a comment on the urban VOC emission uncertainty as follows:

Page 6-Line 13-20: "Urban VOC emissions could be a relevant source for O3 concentration. Over Spanish urban areas, HERMESv2.0 estimates VOC emissions from road transport and the use of solvents (Fig. 1) following bottom-up approaches (Guevara et al., 2013). However, uncertainties in the estimation of urban VOC emission inventories, as stated recently by several works (Pan et al., 2015; Liu et al., 2017; McDonald et al., 2018; Lewis, 2018) makes uncertain the urban VOC contribution to tropospheric O3 concentrations. In order to overcome this problem, continuous monitoring of urban VOC should be performed in Spanish cities, following the example of other regions in which O3 is also a major problem such as Mexico City (Jaimes-Palomera et al., 2016).

In addition, the use of formaldehyde satellite observations to constrain urban VOC emissions could be also pointed out as a future task to improve the representativeness of urban emission inventories (Zhu et al., 2014)"

References: Guevara, M., Martínez, F., Arévalo, G., Gassó, S., and Baldasano, J.M.: An improved system for modelling Spanish emissions: HERMESv2.0, Atmos. Environ., 81, 209–221, doi: 10.1016/j.atmosenv.2013.08.053, 2013.

Jaimes-Palomera, M., Retama, A., Elias-Castro, G., Neria-Hernández, A., Rivera-Hernández, O., and Velasco, E.: Non-methane hydrocarbons in the atmosphere of Mexico City: Results of the 2012 ozone-season campaign, Atmos. Environ., 132, 258–275, https://doi.org/10.1016/j.atmosenv.2016.02.047, 2016

Lewis, A. C.: The changing face of urban air pollution. Science, 359, 744-745. doi:10.1126/science.aar4925, 2018.

Liu, H., Man, H., Cui, H., Wang, Y., Deng, F., Wang, Y., Yang, X., Xiao, Q., Zhang, Q., Ding, Y., and He, K.: An updated emission inventory of vehicular VOCs and IVOCs in China, Atmos. Chem. Phys., 17, 12709-12724, https://doi.org/10.5194/acp-17-12709-2017, 2017

McDonald, B. C., de Gouw, J. A., Gilman, J. B., Jathar, S. H., Akherati, A., Cappa, C. D., Jimenez, J. L., Lee-Taylor, J., Hayes, P. L., McKeen, S. A., Cui, Y. Y., Kim, S.-W., Gentner, D. R., Isaacman-VanWertz, G., Goldstein, A. H., Harley, R. A., Frost, G. J., Roberts, J. M., Ryerson, T. B., and Trainer, M.: Volatile chemical products emerging as largest petrochemical source of urban organic emissions, Science (New York, N.Y.), 359, 760–764, 2018

Soret, A., Guevara, M., Baldasano, J.M., 2014. The potential impacts of electric vehicles on air quality in the urban areas of Barcelona and Madrid (Spain). Atmospheric Environment, 99, 51–63.

Zhu, L., Jacob, D. J., Mickley, L. J., Marais, E. A., Cohan, D. S., Yoshida, Y., Dun-

can, B. N., González Abad, G., and Chance, K. V.: Anthropogenic emissions of highly reactive volatile organic compounds in eastern Texas inferred from oversampling of satellite (OMI) measurements of HCHO columns, Environ. Res. Lett., 9, 114004, doi:10.1088/1748-9326/9/11/114004, 2014.

6. Reviewer #2: It is also widely accepted that biogenic VOCs play a major role on atmospheric photochemistry. For this study, the authors rely on the Model of Emissions of Gases and Aerosols from Nature (MEGAN) version 2.04 (P5.Line 30). This version was revised and extended through version 2.1 (Guenther et al., 2012) that also includes some code fixes. I'd strongly suggest the authors to perform a sensitivity run to understand whether using an outdated version of MEGAN may introduce relevant biases into their simulation. Species tagging and emission categories selection described in section 2.3 seem sensible although VOC emission shares should be reviewed taking into account the previous comment.

Authors: Although we use the MEGANv2.0.4 model, we have used the most updated emission factors (version 2011) from the MEGANv2.1 model (http://lar.wsu.edu/megan/guides.html). In the Section 2 of the supplement we discuss the behaviour of our biogenic emissions using this configuration. Figure S2 shows the isoprene concentration at the Montseny station during the DAURE experimental campaign (Seco et al., 2011; http://cires.colorado.edu/jimenez-group/wiki/index.php/DAURE). This evaluation indicates that modelled isoprene concentrations with updated emission factors are in reasonably good agreement with observations.

We have improved the description of the upgraded MEGAN version used in this study as follows:

Page 6 - Line 9-11: "In this study, we have updated MEGANv2.0.4 with emission factors from last MEGANv2.1 (http://lar.wsu.edu/megan/guides.html). In Sect. 2 of the supplement, we provide a comparison with measurements from the DAURE campaign

[Figure]

(Pandolfi et al., 2014) showing the reasonably good behaviour of our modelled isoprene"

References:

Pandolfi, M., Querol, X., Alastuey, A., Jimenez, J.L., Jorba, O., Day, D., Ortega, A., Cubison, M.J., Comerón, A., Sicard, M., Mohr, C., Prévot, A.S.H., Minguillón, M.C., Pey, J., Baldasano, J.M., Burkhart, J.F., Seco, R., Peñuelas, J., van Drooge, B.L., Artiñano, B., Di Marco, C., Nemitz, E., Schallhart, S.,, Metzger, A., Hansel, A., Lorente, J., Ng, S., Jayne, J., Szidat, S.: Effects of soures and meteorology on particulate matter in the Western Mediterranean Basing: An overview of the DAURE campaign. J. Geophys. Res. Atmos., 119, 4978-5010, 2014

We are currently working on upgrading our modelling system with MEGANv3 (https://bai.ess.uci.edu/megan/versions/megan3).

References: Seco, R., Peñuelas, J., Filella, I., Llusià, J., Molowny-Horas, R., Schallhart, S., Metzger, A., Müller, M., and Hansel, A.: Contrasting winter and summer VOC mixing ratios at a forest site in the Western Mediterranean Basin: the effect of local biogenic emissions, Atmos. Chem. Phys., 11, 13161-13179, doi:10.5194/acp-11-13161-2011, 2011.

7. Reviewer #2: I'm also concerned about a potential double counting and/or erroneous spatio-tempral allocation of VOCs emissions from agriculture since plant functional types considered in MEGAN include crops. From previous literature, a share of 70% of total VOCs from SNAP 11 (nature) seems too high and may support that shortcoming. Please, double check this potential issue since it may bring about a considerable bias the outcomes of the study.

Authors: We are not double counting crops emissions. Emissions from agriculture (SNAP10) only include VOC from manure management and field burning of agricultural residues. We only include VOC emissions from cultivated crops estimated by MEGAN.

We have added a clarification in the revised version of the manuscript as follows:

Page 6 - Line 7-9: "Note that we configured MEGAN to compute VOC emissions from cultivated crops; the agriculture emission module in HERMESv2.0 estimates the VOC from manure management and field burning of agricultural residues."

8. Reviewer #2: The study makes advantage of a remarkably dense network of monitoring sites in the area of study to assess model performance through the computation of a series of common statistics (appendix B). The results however are difficult to interpret in their current form. Please, see corresponding suggestion in the results section.

Authors: The presentation of the evaluation has been improved following the reviewer's suggestions in the results bellow. See Table 2 and Appendix B in the revised version of the manuscript.

9. Reviewer #2: The authors discuss that the episode at hand concentrated an important percentage of exceedances in Spain (if I understood correctly; the first paragraph may be reviewed for the sake of clarity). However, the inspection of panel a) in Fig 2. Does not seem to indicate that this episode was particularly severe since the distribution of MDA8 is not dissimilar to those of previous years, even when they reflect the concentrations over a 6 month period. In addition, the outliers apparently have moderated values. It would be interesting to make the point for such comparison. I'm not completely sure that it is sound to identify a high pollution episode at national level since O3 largely depends on regional features. If O3 levels were actually high all over the modelling domain, the influence of exported ozone (influenced by synoptic conditions) may be too high and thus, the representativeness of the results and the potential implications policy-wise, rather limited. Please, reflect on that.

Authors: Figure 2a supports the reviewer's comment that the selected episode is not the most severe between 2000-2012 that affected all of Spain. However, it comprises a period with high MDA8 O3 concentrations measured at rural background stations, actually the 75th percentile of those values were above the Target Value, similar to the

particularly severe summer of 2003 (Solberg et al., 2008).

This episode is also interesting because it affected Europe (EEE, 2013), for which 33% and 12% of the total number of exceedances were observed for the information threshold and the Target Value in 2012, respectively. The O3 regional context of the episode allows us to study the influence of the imported O3 to Spain.

In the revised version of the manuscript, we have provided a more clear description of the relevance of the episode as follows:

Page 9 - Line 4-23: "Our first estimation of the origin of peak O3 events in Spain focuses on the episode July 21st -31st, 2012. Figure 2a illustrates the relevance of the episode showing the observed MDA8 O3 concentrations trends at the Spanish EIONET stations during the (extended) summers (i.e., from April to September) from 2000 to 2012, together with the concentrations recorded during the episode. Although the selected episode is not the most severe between 2000 and 2012 at national scale, it comprises a period with high MDA8 O3 concentrations measured at rural background stations, actually the 75th percentile of those values were above the Target Value, similar to the particularly severe summer of 2003 (Solberg et al., 2008).

This episode is also interesting because it was widespread and affected big parts of Europe (EEA, 2013). Only during this period 33% and 12% of the total number of exceedances for the information threshold and the Target Value in 2012, respectively, were measured. The O3 regional context of the episode allows us to study the influence of the imported O3 to Spain.

The maps of the 90th percentile of the measured MDA8 O3 concentrations over Spain (Fig. 2b) shows high concentration spots all over the domain. The exceedances of the Target Value were found in the surroundings of large urban areas (Madrid, Barcelona, Valencia, Seville) and along Spanish valleys (i.e., Ebro Valley, Guadalquivir Valley).

There were more than 100 exceedances of the O3 Target Value in most of the days

during the episode, with relative maxima on July 25th, 28th and 31st attributed to the change in the synoptic conditions (Fig. S3). Figure 3 shows the meteorological patterns (temperature at 2m, wind at 10m, precipitation, mean sea level pressure and geopotential height at 500 hPa) modelled by WRF-ARW during the three distinctive days over the outer EU12 domain."

We agree with the reviewer that other O3 episodes should be studied to extract statistically robust conclusions on the main source contribution to O3 events in Spain. To remark that this work is a first quantitative O3 source apportionment study, and the representativeness of our results is limited because they are focused in just one episode, we have added some comments (Section 4) as follows:

Page 20–Line 1-2: "Our study has provided a first estimation of the main sources responsible for high O3 concentration in the Western Mediterranean Basin during the period July 21st -31st, 2012

Page 22–Line 24-25: "[...] future studies should preferentially cover multiple summer periods in order to improve representativeness."

Page 22–Line 22-27: "For regulatory applications, further source apportionment studies should target not only emissions from activity sectors, but also the source regions where the emission abatement strategies should be applied. In addition, future studies should preferentially cover multiple summer periods in order to improve representativeness. We note that our results cannot predict whether emission abatement will have either a positive or a negative effect in O3 changes due to the non-linearity of the O3 generation process. Subsequent source sensitivity analyses tailoring the identified main contribution sources could predict how O3 will respond to reductions in precursor emissions, which are essential to define the most efficient O3 abatement strategies in the Western Mediterranean Basin."

10. Reviewer #2: They also claim that the episode affected the central and north of the IP, but Fig. 2c shows high concentration spots all over the domain (or maybe the

colour scale is not clear enough). It is also hard to see what the influence of Madrid and Barcelona plumes is (something consistent, to my understanding, with dominant stagnant conditions). Please, try to clarify.

Authors: Following the review's recommendation, in the revised version of the manuscript we have improved the colour scale in the aforementioned Figure (now Fig. 2b) to better distinguish the variability of the $O_3$ concentrations. In addition, we have clarified the explanation about the spatial coverage of the episode as follows:

Page 9 - Line 16-18: "The maps of the 90th percentile of the measured MDA8 $O_3$ concentrations over Spain (Fig. 2b) show high concentration spots all over the domain. The exceedances of the Target Value were found in the surroundings of large urban areas (Madrid, Barcelona, Valencia, Seville) and along Spanish valleys (i.e., Ebro Valley, Guadalquivir Valley)."

11. Reviewer #2: Fig 3 illustrates the meteorological conditions through some WRF-ARW outputs. I understand that a thorough model evaluation is not the main purpose of the paper but a minimal check (also through a statistic evaluation) of the credibility of the meteorological simulation (mainly wind fields) may substantially help the authors to gain a better insight of their results. For instance, that would help them to contrast hypotheses such us the one made in P10.Line 25. and following, where they attribute $O_3$ nighttime overestimation to the underestimation of vertical mixing during nighttime stable conditions. In that case, a comparison of observed (where available) and modelled PBLH may be useful.

Authors: According to the reviewer's comment, we have included in Sect. 4 of the supplement an evaluation of wind speed (WS) and direction (WD) at 10 m and temperature at 2 m (T2M) using METeorological Aerodrome Report stations (METAR).

For the selected episode, there were 50 METAR stations located at airports (see location in Fig. S5). Table S2 shows the scores following the methodology explained in "Section 2.4 Evaluation method" for concentrations.

The modelled T2M shows the best behaviour when compared with observations (r=0.91) (Table S2). The model slightly underestimates T2M (-0.2 °C), especially for maximum and minimum temperatures (1.0°C and 0.4 °C for p25 and p75, respectively) (Fig. S5). The model reproduces the WS (r=0.42-0.70) with an overestimation of ∼0.3 ms-1 on average. The overestimation is particularly marked during nighttime (Fig. S5), coincident with low-level wind speeds. These biases may contribute to the underestimation of surface concentrations of O3 precursors. The wind direction shows a lower correlation coefficient (0.1, 0.43).

We did not evaluate the PBL height in this study, but Bank et al. (2012) used daytime radiosounding and PBL height estimations from backscatter lidar to perform a comprehensive evaluation of PBL parametrization from WRF in the North-East Iberian Peninsula. This study found that there is a systematic underestimation of PBL height simulated by WRF. These results are consistent with Vautard et al. (2012), who found that models generally underpredict PBL heights at nighttime.

Overall, nighttime meteorology remains a challenge for meteorological models. The nighttime systematic overestimation of wind and underestimation of PBL height is a potential source of large error compensation for the modelling of NO2 and O3 nighttime concentrations.

The revised version of the manuscript includes a discussion of the meteorological evaluation in the evaluation section as follows:

Page 12 – 18-22: "Section S4 in the supplement discusses the meteorological evaluation results and their impact on the pollutant concentrations. Not surprisingly, temperature shows the best behaviour when compared with observations (Table S2). The modelled wind speed is overestimated, particularly during nighttime (Fig. S5), coincident with low-level wind speed. The nighttime overestimation of wind is a source of error in modelled NO2 and O3 nighttime concentrations (Vautard et al., 2012; Bessagnet et al., 2016)"

References:

Bessagnet, B., Pirovano, G., Mircea, M., Cuvelier, C., Aulinger, A., Calori, G., Ciarelli, G., Manders, A., Stern, R., Tsyro, S., Gar- cía Vivanco, M., Thunis, P., Pay, M.-T., Colette, A., Couvidat, F., Meleux, F., Rouïl, L., Ung, A., Aksoyoglu, S., Baldasano, J. M., Bieser, J., Briganti, G., Cappelletti, A., D'Isidoro, M., Fi- nardi, S., Kranenburg, R., Silibello, C., Carnevale, C., Aas, W., Dupont, J.-C., Fagerli, H., Gonzalez, L., Menut, L., Prévôt, A. S. H., Roberts, P., and White, L.: Presentation of the EURODELTA III intercomparison exercise – evaluation of the chemistry transport models' performance on criteria pollutants and joint analysis with meteorology, Atmos. Chem. Phys., 16, 12667–12701, doi:10.5194/acp-16-12667-2016, 2016.

Vautard, R., Moran, M. D., Solazzo, E., Gilliam, R. C., Matthias, V., Bianconi, R., Chemel, C, Ferreira, J., Geyer, B., Hansen, A.B., Jericevic, A., Prank, M., Segers, A., Silver, J.D., Werhahn, J., Wolke, R., Rao, S.T., Galmarini, S.: Evaluation of the meteorological forcing used for the Air Quality Model Evaluation International Initiative (AQMEII) air quality simulations. Atmospheric Environment, 53, 15-37, 2012.

12. Reviewer #2: As for the discussion of the general meteorological conditions, I'm not sure that the attempt to discriminate between ITL and NWadv situations is nor illustrative or needed. The discussion is hard to follow and the application of deterministic CTM may make that effort redundant.

Authors: Several studies in the Iberian Peninsula (IP) have addressed the causes of O3 episodes looking at the circulation of air masses (Millán, 2014, and references therein). Specifically, recently, some studies found relevant to discriminate by synoptic situations for studying the phenomenology of summer O3 over Catalunya (Querol et al., 2017), Madrid (Querol et al., 2018) and overall in whole Spain (Valverde et al., 2016).

As these authors claim, we believe that distinguishing by synoptic conditions is relevant to understand the O3 origin in Spain. Stagnant conditions are characterized by weak synoptic winds, intense solar radiation, and the development of the Iberian Thermal

Low, which forces the convergence of surface winds from the coast towards the central IP during the day and enhancing mesoscale process, which favours the accumulation of pollutants. In contrast, northwestern advections transport air masses from the Atlantic towards the north and west of the IP, favouring the contribution of imported O3 concentrations.

In addition, distinguishing by synoptic conditions allows assessing the performance of the deterministic model (WRF and CMAQ) to reproduce the synoptic transport and chemistry. For example the time series of source apportionment results in the northwest of the Iberian Peninsula (Fig. 9) show that:

Page 18 – line 16-22: "The time series show that the model reproduces reasonably well the observed O3 variability under different synoptic conditions. O3 reaches the highest concentration (~100/150 $\mu$gm-3 in urban/rural areas) under stagnant conditions (July 24th-27th) when the contribution of anthropogenic sources from all activity sectors is the highest (60-70%). O3 concentrations decrease down to ~70 $\mu$gm-3 under NW advective conditions (e.g., July 28th-30th) when the imported O3 shows the highest contribution (80-90%). Saavedra et al. (2012) found that stationary anticyclones over the NWIP play an important role in the occurrence of high O3 concentrations. Our results show that under these stagnant conditions O3 concentrations are due largely to in situ production (photochemistry) from on-road traffic, shipping, power plants, and industry in almost the same proportion."

References:

Millán, M.M.: Extreme hydrometeorological events and climate change predictions in Europe, J. Hydrol., 518, 206-224, doi: 10.1016/j.jhydrol.2013.12.041,2014.

Querol, X., Gangoiti, G., Mantilla, E., Alastuey, A., Minguillón, M.C., Amato, F., Reche, C., Viana, M., Moreno, T., karanasiou, A., Rivas, I., Pérez, N., Ripoll, A., Brines, M., Ealo, M., Pandolfi, M., Lee, H.-K., Eun, H.-R., Park, Y.-H., Escudero, M., Beddows, D., Harrison, R.H., Bertrand, A., Marchand, N., Lyasota, A., Codina, B., Olid, M.,

Udina, M., Jiménez-Esteve, B., Soler, R.M., Alonso, L., Millán-M., and Ahn, K.-Ho.: Phenomenology of high ozone episodes in NE Spain, Atmos. Chem. Phys., 17, 2817-2838, doi: 10.5194/acp-17-2817-2017, 2017. Querol, X., Alastuey, A., Gangoiti, G., Perez, N., Lee, H. K., Eun, H. R., Park, Y., Mantilla, E., Escudero, M., Titos, G., Alonso, L., Temime-Roussel, B., Marchand, N., Moreta, J. R., Revuelta, M. A., Salvador, P., Artíñano, B., García dos Santos, S., Anguas, M., Notario, A., Saiz-Lopez, A., Harrison, R. M., Millán, M., and Ahn, K.-H.: Phenomenology of summer ozone episodes over the Madrid Metropolitan Area, central Spain, Atmos. Chem. Phys., 18, 6511-6533, https://doi.org/10.5194/acp-18-6511-2018, 2018. Valverde, V., Pay, M.T., and Baldasano, J.M.: Ozone attributed to Madrid and Barcelona on-road transport emissions: characterization of plume dynamics over the Iberian Peninsula, Sci. Total Environ. 543, 670–682, doi: 10.1016/j.scitotenv.2015.11.070, 2016.

13. Reviewer #2: The results of the statistical evaluation for CMAQ outputs are summarized in Table 2. Some suggestions for this table: - Please include in the caption whether the exceedance column refers to observed or modelled values (the missing one may be also included either way) - Two or three decimal points for r may be used - MNB (%) may be more illustrative than MB (that can be derived directly from MM – MO) - It may be misleading to pool together the statistics for different types of monitoring stations. As the authors discuss, it is arguable that outputs from a 4x4 km2 model exercise should be compared against observations at traffic locations. - It is unclear why some monitoring sites wouldn't fit into any of the categories considered. Please elaborate and state the rationale to include them in the analysis. It may be interesting to put these results into perspective by comparing them with those from other modelling exercises based on similar model suites in the IP or elsewhere (besides referring to previous applications of CALIOPE itself).

Authors: The revised version of the manuscript includes all the reviewer's suggestions regarding Table 2.

As the reviewer indicates, it is arguable that outputs from a 4x4 km2 model exercise

should be compared against observations at traffic locations. Actually, traffic station may not be representative of a 4x4 km2 grid. Despite this limitation, we included traffic stations in our analysis discussing the model limitations as follows:

Page 11 – Line 27-28: "Underestimation of NO2 traffic peaks is a common problem in Eulerian mesoscale models (Pay et al., 2014), as emission heterogeneity is lost in the grid cell-averaging process, which is especially critical in urban areas".

The stations without a category corresponded to suburban background (SB) stations. The revised version of the manuscript includes now the SB category in both the discussion and Table 2. Note that now all the stations fit any of the five categories (i.e., IN, TR, UB, SB and RB), so the exceedance/station numbers in the IN/TR/UB/SB/RB rows do sum up to the numbers in the "ALL" rows.

It is difficult to compare the present evaluation results with other modelling studies because of the different period, domain, resolution, model setup, etc. However, we agree with the reviewer that it may be interesting to put these evaluation results into perspective. In this sense, we have added the following paragraph in the revised version of the manuscript:

Page 12 – Line 8-16: "The comparison with previous CALIOPE studies (Baldasano et al., 2011; Pay et al., 2014) indicates that r is in the same range for O3 (0.6-0.7) and NO2 (0.4-0.5) at individual stations; the same applies to RMSE (15-29 $\mu$gO3 m$-3$ and 10-20 $\mu$gNO2 m$-3$). Modelled O3 shows higher performance at traffic stations in large cities, since stations influenced by road transport emissions (i.e., high-NOx environments) are better characterized with a more pronounced daily variability (Baldasano et al., 2011). At European scale, several model intercomparisons (Giornado et al. 2015; Bessagnet et al., 2016) indicate that O3 concentrations in summer agree with the surface observations with r between 0.5 and 0.6. NO2 hourly variably is overall underestimated due to uncertainties in the emission and meteorological modelling and model resolution. These studies highlight the limitations of models to simulate

meteorological variables that affect the NO2 hourly variability, and therefore the model performance for O3 in high-NOx environments and downwind."

References:

Baldasano, J.M., Pay, M.T., Jorba, O., Gassó, S., Jiménez-Guerrero, P., 2011. An annual assessment of air quality with the CALIOPE modeling system over Spain. Sci Total Environ, 409, 2163-2178. Basart, S., Pérez, C., Nickovic, S., Cuevas, E., Baldasano, J.M., 2012. Development and evaluation of the BSC-DREAM8b dust regional model over Northern Africa, the Mediterranean and the Middle East. Tellus B 64, 18539. http://dx.doi.org/10.3402/tellusb.v64i0.18539. Bessagnet, B., Pirovano, G., Mircea, M., Cuvelier, C., Aulinger, A., Calori, G., Ciarelli, G., Manders, A., Stern, R., Tsyro, S., Gar- cía Vivanco, M., Thunis, P, Pay, M.-T., Colette, A., Couvidat, F., Meleux, F., Rouïl, L., Ung, A., Aksoyoglu, S., Baldasano, J. M., Bieser, J., Briganti, G., Cappel- letti, A., D'Isidoro, M., Fi- nardi, S., Kranenburg, R., Silibello, C., Carnevale, C., Aas, W., Dupont, J.-C., Fagerli, H., Gonzalez, L., Menut, L., Prévôt, A. S. H., Roberts, P., and White, L.: Presentation of the EURODELTA III intercomparison exercise – evaluation of the chemistry transport models' performance on criteria pollutants and joint analysis with meteorology, Atmos. Chem. Phys., 16, 12667–12701, doi:10.5194/acp-16-12667-2016, 2016 Giordano, L., Brunner, D., Flemming, J., et al., 2015. Assessment of the MACC reanalysis and its influence as chemical boundary conditions for regional air quality modeling in AQMEII-2. Atmos. Envrion., 115, 371-388. Pay, M.T., Martínez, F., Guevara, M., Baldasano, J.M., 2014. Air quality at kilometre scale grid over Spanish complex terrains. Geosci. Model Dev. 7, 1979–1999. http://dx.doi.org/10.5194/gmd-7-1979-2014.

14. Reviewer #2: P11.L10-14. The description of the performance-based categories is hard to follow and it is already condensed in Fig. 5. Please, simplify or simply remove that passage. I'd suggest the authors to re-compute statistics and assessment with the alternative BVOC emissions model run mentioned earlier.

[Figure]

Authors: We agree with the reviewer. In the revised version of the manuscript, we have removed that passage on the description of the model performance based on bias categories as it is condensed in Fig. 5.

As explained before, we did not perform the BVOC emission sensitivity test with MEGAN comparing v2.0.4 and v2.1, because evaluation of MEGAN with updated emission factors indicates that modelled isoprene concentrations are in reasonably good agreement with observations. We foresee this BVOC emission sensitivity test and its effect on O3 during the upgrade of our modelling system with MEGANv3 (https://bai.ess.uci.edu/megan/versions/megan3).

15. Reviewer #2: If mention to specific cities or areas is made (e.g. Ebro Valley, Lleida Plain), please identify them in any of the maps in the manuscript.

Authors: We have improved Figure 2 to identify the cities and areas that appear throughout the manuscript.

16. Reviewer #2: The information summarized in Fig. 7 is interesting although the concept of "receptor regions" is unclear. It seems reasonable in terms of geographical location but differences regarding contributions from different sectors are not evident, especially if the results are put into the perspective of the typical uncertainties of modelling exercises that can be inferred from the model evaluation previously presented.

Authors: The O3 receptors are defined as air quality stations located in regions with similar meteorological and O3 patterns, main source contributors and geographical patterns. The O3 receptor regions defined in this work are consistent with Diéguez et al. (2014) and Querol et al. (2016), who proposed a similar regionalization based only on observations from air quality stations.

Page 14 – Line 4-6: "We have identified ten O3 receptor regions with similar characteristics in terms of meteorological and geographical patterns, O3 dynamics and main source contributors (Figs. 4 and 6). The receptor regions defined in our work are consistent with Diéguez et al. (2014) and Querol et al. (2016), who proposed a similar regionalization based on observations from air quality stations."

Fig. 7 shows the contribution from different sectors by O3 receptor region ordered by decreasing concentration of imported O3. Differences between sectors are more evident in the normalized contribution (Fig. 7b).

We agree with the referee that the results of the source apportionment have an associated uncertainty. However, this uncertainty cannot be precisely quantified because of the lack of apportioned O3 observations. We note that in sect 3.4 we extensively evaluate and discuss our results using O3, NO2 and wind speed and direction observations. We show that for some cases we can clearly identify a particular sector to be responsible for the O3 mismatch.

17. Reviewer #2: It is interesting noting a relatively large contribution from the SNAP 8 sector. The share of mobile sources is particularly important in the SIP area, which would be consistent with the discussion regarding the influence of shipping. However, other areas such as GV or even CIP present a non-negligible contribution. Could the authors elaborate on that?

Authors: The SNAP8 sector accounts for international shipping, airport service and agricultural machinery. The O3 contribution from non-road transport in the central of the Iberian Peninsula may arise from the international shipping routes, Madrid's airports and the agricultural machinery operating in the surrounding rural areas. The current study has not distinguished these subsectors but it maybe useful for future source apportionment studies. We have clarified the definition of the SNAP8 sector and its contribution in the CIP as follows:

Table 1: "SNAP8: Non-road transport (international shipping, airport and agricultural machinery)"

Page 16- Line 18-20: "The O3 contribution from non-road transport in this region may

arise from the Madrid's airports and the agricultural machinery operating in the surrounding rural areas mainly."

Page 20 – Line 28-29: "The non-road transport sector (including international shipping, airport and agricultural machinery) is as significant as road transport inland (10-19% of the daily mean O3 concentration during the peaks)."

18. Reviewer #2: Maybe the discussion in section 3.4 is too profuse and should be substantially shortened. I encourage the authors to summary here their findings and provide the region-by-region discussion as supplementary material, including Fig. 8 and Fig. 9. Oppositely, the rationale for the station sub-set selection may deserve further explanation. In general, the section is abundant in hypotheses and subjective interpretation that are not clearly supported by evidence. Personally, I don't think this contribution really benefits from such approach. The paper may be restricted to a more solid and consistent analysis and discussion of the findings from the application of CMAQ-ISAM. That is novel and interesting enough and further attempts to relate the results with detailed regional dynamics and atmospheric patterns may very well be addressed in future specific studies (using more specific methods and data, e.g. better resolved emission inventories).

Authors: We have summarized that part following the reviewer's suggestion. Please, have a look at the revised version of the manuscript.

However, we have kept the region-by-region discussion because the main purpose of the present study is the estimation of the contribution of the anthropogenic activity sectors and the imported concentration to peak O3 events in Spain. In addition, this region-by-region discussion has provided a perspective about the potential use of source apportionment analysis for improving the O3 modelling and designing future mitigation strategies at regions with a high on-road traffic contribution (i.e., CIP and NEIP in Fig. 8) and a high contribution from industry and energy production (i.e., NWIP and Guadalquivir Valley in Fig. 9).

19. Reviewer #2: The authors claim that the modelling exercise presented allowed an in-depth evaluation of the modelling system applied. This relates to my last comment regarding the results section. The paper may lack a well-defined objective and presents a huge amount of information without a clear purpose. For example, if the main interest was to assess the modelling system capabilities and identify options for improvement, the results and analysis should gravitate towards a more detailed statistical analysis within a better defined methodological framework. I acknowledge a valuable study but I think the authors should revise their manuscript under a clearly defined scientific question avoiding an excessive spread in their discussion that may lead to inconsequential or cursory analyses and reflect that also in this section. As for the discussion on model uncertainty I find particularly important to take into account the observations regarding biogenic emissions, although the mismatch between emissions and meteorological conditions may also hinder the discriminating power of the results.

Authors: We have rewritten the objective in the abstract and the introduction to clarify the objective of the study as follows:

In the abstract (Page 1 – Line 19-21): "The main goal of this study is to provide a first quantitative estimation of the contribution of the main anthropogenic activity sectors to peak $O_3$ events in Spain relative to the contribution of imported (regional and hemispheric) $O_3$. We also assess the potential of our source apportionment method to improve $O_3$ modelling"

In the introduction (Page 4 – Line 25-30): "The integrated source apportionment tools combined with high-resolution emission and meteorological models can help unravelling the sources responsible for peak summer events of $O_3$ in the Western Mediterranean Basin. Quantifying the contribution of emission sources during acute $O_3$ episodes is a prerequisite for the design of future mitigation strategies in the region. In this framework, the main goal of this study is to provide a first quantitative estimation of the contribution of the main anthropogenic activity sectors compared to the imported concentration (regional and hemispheric) to peak $O_3$ events in Spain. We also assess

the potential of our source apportionment method to improve O3 modelling"

Under clearly defined scientific objectives, we have substantially shortened Section 4 to avoid an excessive spread in the discussion. A new version of Section 4 is available in the revised manuscript.

As suggested by the reviewer we have included a comment on the uncertainty of biogenic emissions and emissions reference year as follows:

Page 22 – Line 10-12: "Another relevant and uncertain source for O3 concentration is the urban VOC emissions. Future research works should be devoted to continuous monitoring of urban VOC and take advantage of satellite observations to improve speciation and spatial variability of urban VOC emissions."

20. Reviewer #2: In any case, caution should apply since the timespan of the period analysed makes it difficult to extract general conclusions. This is particularly important for the regulatory implications that may be derived from this study. As the authors conclude, I find reasonable to base recommendations for abatement strategies in more specific, regional scale, detailed analyses. Consequently, I'd keep such conclusions to a minimum in this contribution.

Authors: As discussed before, we agree that the representativeness of our results is limited because they are focused in just one episode. Future studies should preferentially cover multiple summer periods in order to improve representativeness.

Although, the main goal of this study is not the design abatement policies, these source apportionment results has provide a perspective about the potential use of these methods for regulatory studies in non-attainment regions as a prerequisite for the design of future mitigation strategies. We have added a short comment on this in the discussion section:

Page 23-Line 4-7: "This work has quantified the local and imported contributions to O3 during an episode in a particular area in southwestern Europe. In addition, we

have provided a perspective about the potential use of source apportionment method for regulatory studies in non-attainment regions. Further O3 source apportionment studies targeting other nonattainment regions in Europe are necessary prior to design local mitigation measures that complement national and European-wide abatement efforts."

21. Reviewer #2: Please revise equations 1 to 4 for a better readability

Authors: We have increased the size of the font in the equation for better readability.

22. Reviewer #2: P7.Line 4: SNAP3 and 4

Authors: We note that SNAP34 is not a typo error. We have defined sector SNAP34 all together as mentioned in Table 1 as manufacturing industries. We follow the same reporting approach as the one proposed by the TNO_MACC emission inventories, in which SNAP 3 and SNAP 4 emissions are merged to SNAP 34. This does not mean that we deal with SNAP3 and SNAP4 emissions all together. Emissions from each point source are estimated individually and applying specific activity and emission factors, as well as speciation and temporal profiles. It is just a matter of reporting format.

23. Reviewer #2: P8.Lines 12-13: the brackets are not needed: "In Spain, around 60% of the annual exceedances also occurred during this period. (As shown in Querol et al. (2016) July is typically the month with the highest number of O3 exceedances in Spain.) The" Authors: We have removed this statement for simplicity.

24. Reviewer #2: P11.Line 11: "for 93% of the stations" instead of "for the 93% of stations" Authors: We have amended this issue in the revised version of the manuscript.

25. Reviewer #2: P12.Line 2: "extremely low winds"? Authors: We have rewritten this statement as follows "stagnant conditions".

26. Reviewer #2: P20.Line 8: "O3at" is missing a space Authors: We have amended this issue in the revised version of the manuscript.
27. Reviewer #2: P34.Line 5 (Fig. 2 caption): I guess the authors mean "Number of stations" instead of "Number of days" Authors: we have rewritten the caption of Figure 2 following the reviewer's suggestions as follows:

Figure 2-caption: "Number of the Spanish EIONET stations days exceeding the O3 Target Value (120 $\mu$g/m3) per episode day"

Please also note the supplement to this comment:
https://www.atmos-chem-phys-discuss.net/acp-2018-727/acp-2018-727-AC1-supplement.zip

---

## Author Comment (AC2) · 25 Feb 2019

1. Reviewer #3: The paper gives important contribution to address the source apportionment study regarding ozone episode occurred in Spain. The paper is well structured and presents a complete analysis of the modelling results. However, there are some major points that should be addressed before recommended for publication. Besides that, English should be revised along the manuscript, there is some inconsistencies and grammatical errors. See below major and minor comments.

Authors: We thank the reviewer #3 for the comments and suggestions for improvement. We have corrected errors and omissions, and introduced as much as possible the reviewer's suggestions.
Please, find below the item-by-item response. For more details on the review process, we have uploaded the manuscript with track-changes.

2. Reviewer #3: Abstract; Line 15 (Page 4/Line 2): there is recently studies that showed that source-apportionment methods are not adequate to investigate plans and mitigation measures, in particular for non-linear pollutants like ozone, and that for that purpose “scenario analysis” based on “brute-force” are recommended. Authors should revise the text along the manuscript where it is mentioned the purpose of “designing plans”, which should not be the final objective of this source-apportionment study. (see Clappier, A., Belis, C., Pernigotti, D., Thunis, P. Source apportionment and sensitivity analysis: two methodologies with two different purposes. Geosci. Model Dev. Discuss. 10, 4245-4256 (2017))

Authors: We totally agree with the reviewer’s point of view and with the content of Clappier et al., (2017). The main goal of this study is not the design abatement policies, but it is to provide a first quantitative estimation of the contribution of the main anthropogenic activity sectors to peak O3 events in Spain relative to the contribution of imported O3.

Actually, source apportionment techniques alone cannot be used to the design abatement policies. Subsequent source sensitivity analyses tailoring the identified main contribution sources could predict how O3 will respond to reductions in precursor emissions. Both, source apportionment and source sensitivity are complementary and essential studies to define the most efficient O3 abatement strategies in the Western Mediterranean Basin. The manuscript highlights now this idea in different sections that we recap as follows:

Page 4 – Line 26-30: “Quantifying the contribution of emission sources during acute O3 episodes is a prerequisite for the design of future mitigation strategies in the region. In this framework, the main goal of this study is to provide a first quantitative estimation of the contribution of the main anthropogenic activity sectors compared to the imported

concentration (regional and hemispheric) to peak O3 events in Spain."

Page 22–Line 23-27: "We note that our results cannot predict whether emission abatement will have either a positive or negative effect in O3 changes due to the non-linearity of the O3 generation process. Subsequent source sensitivity analyses tailoring the identified main contribution sources could predict how O3 will respond to reductions in precursor emissions, which are essential to define the most efficient O3 abatement strategies in the Western Mediterranean Basin."

Page 23-Line 4-7: "This work has quantified the local and imported contributions to O3 during an episode in a particular area in sourthwestern Europe. In addition, we have provided a perspective about the potential use of source apportionment method for regulatory studies in non-attainment regions. Further O3 source apportionment studies targeting other nonattainment regions in Europe are necessary prior to design local mitigation measures that complement national and European-wide abatement efforts."

In order to be clear in the source apportionment applications and limitations of certain methods, we have added in the revised version of the manuscript a comment taking into account the recent findings in Clappier et al. (2017) as follows:

Page 4-Line 7-11: "Brute force is simple to implement, as it does not require additional coding in the CTM. However, as it quantifies the contribution of each source based on its absence, it does not reproduce actual atmospheric conditions, and therefore it is susceptible to inaccuracies in the prediction of O3 peaks under non-linear regimes (Cohan and Napelenok, 2011). Actually, brute force is not suitable to retrieve source contribution when the relationship between emissions and concentration is non-linear, but it is useful for analysing the concentration responses to emission abatement scenarios (Clappier et al., 2017)."

3. Reviewer #3: Page 4/Line 7-9: please review this sentence according to what has been commented before.

Authors: The statement in Page 4/Line 7-9 has been expanded following the reviewer's suggestion as shown in the previous authors' answer (see Page 4-Line 9-13).

4. Reviewer #3: Page 5/Line 26: the authors should comment about the representativeness of the 2009 emissions to the 2012 SA study presented. From 2009 to 2012 several changes happened in society and economy which was reflected in emission data.

Authors: Our methodological choice has been to use a detailed bottom-up emission inventory instead of a typical top-down regional emission inventory. Bottom-up emissions, estimated using source-specific emission factors and activity statistics, accurately characterise pollutant sources and allow obtaining more realistic results than the ones reported by top-down or regional emission inventories. However, they require very large efforts to be compiled and consequently the updating processes cannot be implemented year-to-year.

In HERMESv2 emissions are based on 2009 data, which was the closest year with updated information on local emission activities at the time this work started.

To understand the impact of the use of 2009 data to study year 2012 we revised the EMEP Centre on Emission Inventories and Projections (EMEP-CEIP), which collects and reviews the national emission inventories from Parties to the Convention on Long-range Transboundary Air Pollution (CLRTAP). Between 2009 and 2012 total NOx and NMVOC emissions in Spain decreased by -10.6% and -10.7%, respectively (EMEP CEIP, 2019). For NOx, around 80% of this reduction is linked to a reduction of road transport emissions, whereas in the case of NMVOC $\sim$50% of the reduction is due to a decrease of industrial emissions. NOx emissions from shipping in Europe have also decreased in the period 2009-2012 by 15%.

For our modelling study, we consider these differences as small and acceptable, and not creating any major inconsistency. The difference of 10-15 % in emissions for certain precursors between 2009 and 2012 is within the typically larger ranges of uncertainty

in emission inventories. We also note that all our results are thoroughly evaluated and critically assessed using observations.

In any case, we have followed the reviewer's suggestion, and we have discussed in the manuscript the potential impact of these differences when the contribution of each emission sector is analysed:

Page 17–Line 32-33: "[. . .] This factor, added to the 15% decrease of NOx shipping emissions observed in Europe between 2009 (HERMESv2.0 base year) and 2012 (EMEP CEIP, 2019) could explain the discrepancies observed."

Page 18–Line 12-14: "[. . .] it has been estimated that between 2009 and 2012 energy production in coal-fired power plants increased from 13.1% to 19.4% (UNESA, 2012), which implied an increase of NOx emissions from the power industry sector of around 19.5% (EMEP CEIP, 2019)."

The Section 4 of the revised version of the manuscript includes now a comment on the methodological implication of using 2009 emissions for O3 source apportionment studies in an episode in 2012 as follows:

Page 5-Line 31-32: "HERMESv2.0 is currently based on 2009 data, which is the closest year with updated information on local emission activities at the time this work started."

Page 21-Line 31: "Our methodological choice has been to use a detailed bottom-up emission inventory instead of a typical top-down regional emission inventory. Bottom-up emissions, estimated using source-specific emission factors and activity statistics, accurately characterise pollutant sources and allow obtaining more realistic results than the ones reported by top-down or regional emission inventories. To understand the impact of the use of 2009 data to study year 2012, we revised the EMEP Centre on Emission Inventories and Projections (EMEP-CEIP), which collects and reviews the national emission inventories from Parties to the Convention on Long-range Transboundary Air Pollution. Between 2009 and 2012, total NOx and NMVOC emissions in Spain decreased by -10.6% and -10.7%, respectively (EMEP CEIP, 2019). For NOx, around 80% of this reduction is linked to a reduction of road transport emissions, whereas in the case of NMVOC ∼50% of the reduction is due to a decrease of industrial emissions. For our modelling study, we consider these differences as small and acceptable, and not creating any major inconsistency. The difference of 10-15 % in emissions for certain precursors between 2009 and 2012 is within the typically larger ranges of uncertainty in emission inventories."

Reference:

EMEP CEIP, 2019. Officially reported emission data. Available at: http://www.ceip.at/ms/ceip_home1/ceip_home/data_viewers/official_tableau/ (last access February 2019)

5. Reviewer #3: Page 7, Line 9: SNAP2 activity can be a particular important source for ozone precursors. Authors should comment about that when they mentioned that SNAP2 is aggregated with other activity sectors.

Authors: One limitation of the current version of CMAQ-ISAMv5.0.2 is that the number of tagged sources increases the computational time. In the current version of CMAQv5.0.2 the increase of the computational resources does not decrease the computational time. A more computationally efficient version of the ISAM will be released with the final version of CMAQv5.3 in Spring 2019. Based on that limitation, we configured our first study tagging the energy, industrial, road transport and non-road transport sectors (Fig. 1b), which account for 92% of the total NOx emissions in Spain. The remaining emission sectors are lumped in the OTHER tag. This selection criterion is explained in the manuscript as follows:

Page 7- Line 20-23: "The selected (tagged) SNAP categories in this study are the energy, industrial, road transport and non-road transport sectors (Fig. 1b), which account for the 92% of the total NOx emissions in Spain. An additional tracer (OTHR) gathers the remaining emission categories that were not explicitly tracked (i.e., SNAP2, 5, 6, 9,

10 and 11).”

We have seen in other studies performed over different domains (e.g., Portugal) that SNAP2 can be an important contributor to O3 precursors. We have mentioned this fact in the manuscript as follows:

From Page 13- Line 33 to Page 14-Line 2: “The high OTHR concentration around the biggest cities in Portugal may be related to precursors emitted by the residential sector (SNAP2 and 9) and biogenic emissions, as found in other source apportionment studies over Portugal (Borrego et al., 2016; Karamchandani et al., 2017).”

6. Reviewer #3: Page 7, Lines 25-30: in the scope of FAIRMODE – Forum for Air Quality Modelling in Europe – tools were developed, namely the DELTA-Tool, to evaluate air quality models and conclude about their suitability to be used for legislation purposes. The authors should consider the application of this tool to evaluate model performance instead of calculating the traditional statistical indicators. In any case, authors should justify why they decided not using this tool.

Authors: The evaluation with the Delta tool has been taken into account in previous evaluation studies of the CALIOPE system, although it has not been shown in the manuscript. We have added this comment to complement this information:

Page 11-Line 2-6: “CALIOPE has been evaluated in detail elsewhere (Pay et al., 2014 and references therein). Furthermore, the system has been evaluated using the tool developed by the Forum for Air Quality Modelling in Europe, named DELTA-Tool (Thunis and Cuvelier, 2014) to support and harmonize the model evaluation in the frame of the Air Quality Directive. Valverde et al. (2016a; 2016b) used the DELTA-Tool v4.0 and showed that the CALIOPE system accomplishes the quality objectives as defined in the Air Quality Directive for 78% to 91% of the NO2 and O3 monitoring stations in summer conditions in 2012.”

References: Thunis, P., Cuvelier, C., 2014. DELTA Version 4.0. Joint Research Center, Ispra (http://aqm.jrc.ec.europa.eu/DELTA/assessment/data/DELTA_UserGuide_V4_0.pdf).

Valverde, V., Pay, M.T., and Baldasano, J.M.: Ozone attributed to Madrid and Barcelona on-road transport emissions: characterization of plume dynamics over the Iberian Peninsula, Sci. Total Environ. 543, 670–682, doi: 10.1016/j.scitotenv.2015.11.070, 2016a. Valverde, V., Pay, M.T., and Baldasano, J.M.: A model-based analysis of SO and NO dynamics from coal-fired power plants under representative synoptic circulation types over the Iberian Peninsula, Sci. Total Environ., 541, 701-713, doi: 10.1016/j.scitotenv.2015.09.111, 2016b.

7. Review #3: Page 13, Lines 16-17: this information (model performance in terms of o3 peaks) should be presented and discussed in the model validation section.

Authors: The information provided in these lines is not a dedicated evaluation analysis on O3 peaks. It is devoted to provide a general perspective on source apportionment results shown in Figure 7. In order to make it clear, we have rewritten this sentence as follows:

Page 14-Line 15-16: “Figure 7 indicates that during exceedances of the MDA8 target value there is a good agreement (r = 0.79) between the sum of apportioned O3 and the observed concentrations over the receptor regions.”

8. Reviewer #3: Page 13, Line 22: this sentence should be completed with information about the area where this impact (up to 8%) is verified.

Authors: The reviewer is right. We have rewritten the sentence as follows:

Page 14 – Line 20-21: “Shipping emissions in the MED region contributed up to 8% of the total O3”.

9. Reviewer #3: Page 15, Lines 4-5: The authors should quantify how “model reproduces reasonably well”

Authors: We agree with the reviewer and we have quantified the model performance at this station as follows:

Page 15-Line 33: “At the urban station, the model reproduces the O3 traffic cycle (r = 0.66 and MB=22.5 $\mu$gm$-$3) featuring the typical low O3 concentrations (< 40 $\mu$gm-3) in the early morning and in the afternoon due to O3 titration (Fig. 8a).”

10. Reviewer #3: Page 15, Lines 9-10: Please clarify the sentence “The NO2 overestimation correlates with the highest road transport contribution”. Page 15, Lines 11-12: Please explain why: “The results point towards a poor representation of the vertical mixing during the stagnant conditions”

Authors: In order to clarify this sentence, we have rewritten this paragraph as follows:

Page 16-Line 2-5:”O3 was overestimated (MB type D) during daytime peaks due to the overestimation of the NO2 morning peaks during stagnant conditions, coincident with the highest road transport contribution for both pollutants. The results point towards a poor representation of the meteorological condition in the city during the stagnant conditions as shown in the meteorological evaluation (Sect. 4 in the supplement)”

11. Reviewer #3: Page 16, Lines 4-10: this should be placed in the model validation section.

Authors: We agree with the reviewer that evaluation results should be addressed in the dedicated section. However, the interpretation of the source apportionment results benefits from model evaluation, and at the same time the source apportionment results support enhanced model evaluation. We have added a comment on this in the Section 2.4 Evaluation method to clarify this issue as follow:

Page 8-Line 24-27: “Evaluation results are discussed together with the source apportionment results. On the one side, the interpretation of the source apportionment results benefits from model evaluation. On the other side, the source apportionment results support enhanced model evaluation as it allows identifying potential errors in

emission estimates for specific sectors and/or in the chemical boundary conditions."

12. Reviewer #3: Page 16, Lines 13-14: how can the authors conclude that the model is able to reproduce all these different processes? Can the authors support better this statement?

Authors: As shown in Fig. 8d, modelled O3 peaks (> 120 $\mu$gm$-$3) are in a good agreement with observations (Fig. 8d), which suggests that overall the model reproduces the main transport paths, photochemical processes, and relative contributions from different sources. We have rewritten this statement as follows:

Page 16-Line 32: "At the rural station, modelled O3 peaks (> 120 $\mu$gm$-$3) are in a good agreement with observations (Fig. 8d), which suggests that overall the model reproduces the main transport paths, photochemical processes, and relative contributions from different sources."

13. Reviewer #3: Page 18, Line 4: since only one rural station is analysed, the authors should not generalize as "In rural background areas"

Authors: The reviewer is right; we have rewritten the sentence accordingly:

Page 18-Line 8-9: "Despite the O3 biases during stagnant conditions, the modelled O3 concentration is in general agreement with observations at the rural background station (Fig. 9d)"

14. Reviewer #3: Page 18, Lines 30-33 to Page 19: the authors analyse the vertical profile in a single point, but this will be not representativeness of the all study domain. Authors should change the text according to this limitation and comment it, or increase the number of points analysed.

Authors: The Figure 10 does not show a vertical profile, but it shows the O3 distribution over a vertical cross section crossing the IP from the west to the east at the centre of the domain (approximately at a latitude of 40.38° N). Although Figure 10 is not representative of the whole domain of study, it helps to understand the vertical variability

of both pollutants with the PBL dynamics as schematized by Millán et al. (1996). We have clarified this point in the manuscript as follows:

Page 19-Line 1-3: “Figure 10 shows the vertical cross-sections at 6, 12, and 18 UTC for O3 and NO2 at a constant latitude (40.38° N) on July 25th, 28th and 30th. It helps to understand the vertical variability of both pollutants according to the dynamics of the PBL as schematized by Millán et al. (1996).”

15. Reviewer #3: Figure 2: please review the figure caption “Number of days exceeding the O3 target value (120 ug.m-3) by each day of the episode”

Authors: We have rewritten the caption of Figure 2 following the reviewer’s suggestions as follows: Figure 2-caption: “Number of stations exceeding the O3 Target Value (120 $\mu$g/m3) per episode day”

Note that in the reviewed version this figure corresponds to the Figure S3 in the Supplement.

16. Review #3: Page 1/Line 20: write 4x4 km2 instead of 4x4 km (please correct this along the manuscript)

Authors: We have amended that.

17. Reviewer #3: Authors should refer the modelling system (CALIOPE) in the abstract.

Authors: The suggestion of the reviewer has been included in the new version of the manuscript as follows:

Abstract (Page 1-Line 21-24): “Our study applies and thoroughly evaluates a countrywide O3 source apportionment method implemented in the CALIOPE air quality forecast system for Spain at high resolution (4 x 4 km2) to understand and quantify the origin of peak O3 events over a 10-day period covering the most frequent synoptic summer conditions in the Iberian Peninsula.”

18. Reviewer #3: Page 3, Line 4: The following reference should be added, since it is

the biggest ozone episode occurred in IP region: “Monteiro A., Gama C., Candido M., Ribeiro I., Lopes M. (2016) Investigating ozone high levels and the role of sea breeze on its transport. Atmospheric Pollution Research 7, 339-347.

Authors: We have added Monteiro et al. (2016) to the list of studies as an example of a large ozone episode occurred in IP region.

19. Reviewer #3: Page 7, Line 23: please indicate how many stations measure both O3 and NO2 pollutants.

Authors: We have mentioned this information latter on when we introduce the NO2/O3 source apportionment time series:

Page 8-Line 20-22: “This work will only discuss in detail the source apportionment plots at key O3 receptor regions, given the high number of stations (260) that simultaneously measure O3 and NO2.”

20. Reviewer #3: Page 10, Lines 17, 24, 29: please add “average” when mentioning “hourly O3” (the values presented are an average of different locations and not an “hourly O3 data”

Authors: The suggestion of adding “average” when mentioning “hourly concentration” has been implemented in the whole manuscript.

21. Reviewer #3:- Page 12, Lines 2-7: the following reference should be added to support this part: Borrego C., Monteiro A., Martins H., Ferreira J., Fernandes A.P., Rafael S., Miranda A.I., Guevara M., Baldasano J.M. (2016). Air quality plan for ozone: a case-study for North Portugal. Air Quality, Atmosphere & Health 9 (5), 447–460.

Authors: The reviewed version of the manuscript includes the reference Borrego et al., (2016) over northern Portugal to support the fact that, under stagnant conditions, imported O3 is depleted and O3 photochemical production is enhanced around the largest industrial/urban areas. Page 13-Line 4-6: “In a source attribution study over northern Portugal, Borrego et al. (2016) also found a reduction of imported O3 and the

subsequent O3 formation by local sources under similar meteorological conditions."

22. Reviewer #3: Page 14, Line 30: please review the English.

Authors: The reviewed version of the manuscript has rewritten this paragraph as follows:

Page 15-Line 28-29: "The following sections analyse the source apportionment results at key regions (see Fig. 5) with a high on-road traffic contribution (i.e., CIP and NEIP) and a high contribution from industry and energy production (i.e., NWIP and Guadalquivir Valley)."

23. Reviewer #3: Page 17, Lines 19-24: authors should consult and use the following reference that compares the different shipping emission inventories mentioned: Russo M.A., Leitão J., Gama C., Ferreira J., Borrego C., Monteiro A. (2018) Shipping emissions over Europe: a state-of-the-art and comparative analysis. Atmospheric Environment 177, 187–194.

Authors: The reviewed version of the manuscript include the reference Russo et al. (2018) to discuss the uncertainties on shipping emissions over Europe:

Page 17-Line 28-32: "A recent review on the state-of-the-art of marine traffic emissions (Russo et al., 2018) indicates that STEAM appears as the most reliable and detailed emissions inventory since it is based on Automatic Identification System data and specific vessel information, with a resolution of 2.5 x 2.5 km2 (Jalkanen et al., 2016). A comparative analysis indicates that EMEP gridded inventories are overestimated, in particular over hotspots in the Mediterranean shipping routes, and underestimated in secondary routes."

References:

Jalkanen, J.-P., Johansson, L., and Kukkonen, J.: A comprehensive inventory of ship traffic exhaust emissions in the European sea areas in 2011, Atmos. Chem. Phys., 16, 71-84, https://doi.org/10.5194/acp-16-71-2016, 2016.

Russo, M. A., Leitão, J., Gama, C., Ferreira, J., & Monteiro, A.: Shipping emissions over Europe: A state-of-the-art and comparative analysis. Atmospheric Environment, 177, 187-194, 2018.

24. Reviewer #3: Page 18, Line 22: please replace "These O3 ..." by "The results presented before ... "

Authors: We have amended that the suggestion.

25. Reviewer #3: Figure 3: Please review the units used along the manuscript, like "m/s"

Authors: We have harmonize the use of units as ms-1.

Please also note the supplement to this comment:
https://www.atmos-chem-phys-discuss.net/acp-2018-727/acp-2018-727-AC2-supplement.zip